# Spatiotemporal integration of contextual and sensory information within the cortical hierarchy in human pain experience

Suhwan Gim [1,2,3], Seok-Jun Hong [1,2,3,4,5], Elizabeth A. Reynolds Losin [6,7], Choong-Wan Woo [1,2,3,5] *

**1** Center for Neuroscience Imaging Research, Institute for Basic Science, Suwon, South Korea, **2** Department of Biomedical Engineering, Sungkyunkwan University, Suwon, South Korea, **3** Department of Intelligent Precision Healthcare Convergence, Sungkyunkwan University, Suwon, South Korea, **4** Center for the Developing Brain, Child Mind Institute, New York, New York State, United States of America, **5** Life-inspired Neural Network for Prediction and Optimization Research Group, Suwon, South Korea, **6** Department of Psychology, University of Miami, Coral Gables, Florida, United States of America, **7** Department of Biobehavioral Health, Pennsylvania State University, University Park, Pennsylvania, United States of America

* waniwoo@skku.edu

**Data Availability Statement:** All relevant data used to generate figures, as well as the scripts for analysis, are available in the Figshare repository at

## Abstract

Pain is not a mere reflection of noxious input. Rather, it is constructed through the dynamic integration of current predictions with incoming sensory input. However, the temporal dynamics of the behavioral and neural processes underpinning this integration remain elusive. In the current study involving 59 human participants, we identified a series of brain mediators that integrated cue-induced expectations with noxious inputs into ongoing pain predictions using a semicircular scale designed to capture rating trajectories. Temporal mediation analysis revealed that during the early-to-mid stages of integration, the frontoparietal and dorsal attention network regions, such as the lateral prefrontal, premotor, and parietal cortex, mediated the cue effects. Conversely, during the mid-to-late stages of integration, the somatomotor network regions mediated the effects of stimulus intensity, suggesting that the integration occurs along the cortical hierarchy from the association to sensorimotor brain systems. Our findings advance the understanding of how the brain integrates contextual and sensory information into pain experience over time.

## Introduction

Pain experience can be understood as a continuous integration process of incoming noxious input with contextual information, inferring the current state of both external and internal environments [1–3]. For example, the impact of expectation on pain perception, such as placebo and nocebo effects, can be understood as a dynamic process, in which the brain continuously integrates sensory inputs with contextual information [4,5], shaping the perception by consistently forming and updating expectations (or prediction) [6]. Previous studies support this concept, suggesting pain perception as a continuous integration and prediction process

https://doi.org/10.6084/m9.figshare.25040951 and Supporting information (S1 Data).

**Funding:** This work was supported by IBS-R015-D1 (Institute for Basic Science; https://www.ibs.re.kr/eng.do; to C.-W.W.), 2021M3E5D2A01022515 (National Research Foundation of Korea; https://www.nrf.re.kr/eng/main; to C.-W.W.), and HI19C1328 (Korea Health Technology R&D Project through the Korea Health Industry Development Institute; https://www.khidi.or.kr/eps, the Ministry of Health & Welfare, Republic of Korea; https://www.mohw.go.kr/eng/; to S.G.). The funders had no role in designing the study, collecting and analyzing the data, deciding to publish, or preparing the manuscript.

**Competing interests:** The authors have declared that no competing interests exist.

**Abbreviations:** aINS, anterior insular cortex; CI, confidence interval; EPI, echo-planar image; FDR, false discovery rate; FIR, finite impulse response; fMRI, functional magnetic resonance imaging; FSL, FMRIB Software Library; FWHM, full width at half maximum; GLM, general linear model; gLMS, generalized labeled magnitude scale; ICA, independent component analysis; IPS, intraparietal sulcus; ITL, inferior temporal lobe; MDS, multidimensional scaling; PCC, posterior cingulate cortex; ROI, region of interest; SBRef, single-band reference; SMA, supplementary motor area; SMC, somatomotor cortex; SD, standard deviation; TP, temporal pole; VIF, variance inflation factor; vlPFC, ventrolateral prefrontal cortex.

across different levels of analysis, from behavioral [7,8] and physiological [9] to neural levels [9,10]. Moreover, there has recently been an increasing emphasis on examining perception, action, and decision-making as a continuous dynamical process [11,12]. However, many previous studies have examined brain mechanisms of the integration process by studying each process separately—i.e., pain anticipation, stimulus inputs, and pain ratings—as "individuated elements" [9–11,13,14]. To address this gap, our current study investigates the temporal dynamics of brain mediation that underpins the continuous integration process of pain perception.

In this study, we employed temporal mediation analysis to examine the intricate interplay between pain modulatory factors, brain activities, and behavioral outcomes. Mediation analysis is a statistical method to assess whether the relationship between an independent variable ($x$; cue or stimulus intensity) and a dependent variable ($y$; ratings) is explained by a mediator ($m$; brain activity), which must significantly reduce or eliminate the direct effect of $x$ on $y$ when included in the model, indicating mediation [15,16]. Previous studies have successfully utilized mediation analysis to identify brain regions that mediate the pain modulation effects of cue-induced expectancy and the intensity of noxious stimuli [13,15–17]. For example, the primary and secondary somatosensory cortices (S1 and S2), along with the dorsal posterior insular cortex (dpINS) have been shown to mediate the effects of stimulus intensity on pain perception [17]. The ventrolateral prefrontal cortex (vlPFC) and anterior insular cortex (aINS) have been identified as mediators of cue-induced pain modulation [15]. Furthermore, the frontoparietal and dorsal attention brain networks have been implicated in mediating the effects of social information on pain perception [13]. However, these studies did not focus on the temporal dynamics of the mediation effects, thereby limiting their ability to examine pain as a continuous integration process. To overcome this limitation, we utilized temporal mediation analysis, a method designed to identify the time-series of brain mediators throughout the trial.

The application of this method faces limitations with conventional experimental paradigms that largely rely on 'button-based' or 'single value-based' behavioral assessments [9,10,13–15,17], given that they are typically optimized to probe segmented processes [11]. For example, the conventional pain rating task typically asks participants to report their overall pain intensity or unpleasantness as a single value, thus treating the ongoing pain process as a single and segmented event and overlooking its inherently dynamic nature. Furthermore, in conventional continuous rating scales in functional magnetic resonance imaging (fMRI) studies, rating-related motor responses are often confounded with pain intensity [18], as higher pain intensity requires further movement, e.g., to the right [19,20]. To address these issues, we devised a semicircular rating scale that places the starting point equidistant from all possible ratings, minimizing the effect of movement distance on ratings. This allows us to implement a "trajectory-based" assessment. The trajectory-based measurements have proven effective in capturing real-time temporal dynamics of motor or cognitive processes in multiple contexts [12,21–26]. The trajectory-based rating method combined with temporal mediation analysis offers a novel avenue to scrutinize the details of a dynamic process of how the brain integrates sensory inputs with prior contextual information over time.

The primary research questions of the current study are as follows. (1) Can we reveal the continuous integration process underlying pain perception with the trajectory-based measurement? and (2) Can we identify the specific timing and brain regions that mediate the temporal dynamics of this integration process? We addressed these questions using an fMRI experiment using a cue-induced pain modulation task. Specifically, participants were asked to continuously report their real-time prediction of pain intensity on the semicircular rating scale using a joystick while receiving a gradually changing heat stimulus. Pain expectation was induced

using pain-predictive cues based on social information, represented as dots on the rating scale before heat stimulation. The rating trajectories were operationalized as an indicator of a real-time pain integration process [11,12,24]. Using this trajectory data, we conducted multilevel general linear models to examine the ongoing integration process. Then, employing temporal mediation analysis combined with a data-driven dimensionality reduction method, we identified spatiotemporal patterns of brain mediators of the integration process of cues and stimulus intensity over time.

Our hypotheses primarily focus on the potential roles of the lateral prefrontal cortices in encoding contextual information [27–32]. Furthermore, we anticipated that, during this integration process, brain regions along the cortical hierarchy (or the association-sensorimotor axis) [33–36] may exhibit distinct temporal mediation profiles for the cue and stimulus effects. For example, association brain regions could play an important role in mediating the effects of contextual cue information, whereas sensorimotor brain regions may primarily mediate the stimulus effects on pain. Overall, this study seeks to elucidate when and which brain regions and networks contribute to pain construction at different temporal stages.

## Results

### Behavioral results

Inside the scanner, participants performed a cue-induced pain modulation task involving continuous pain prediction ratings. Participants were instructed to continuously report their ongoing pain predictions in response to the question, "How much pain do you predict?", using the semicircular rating scale (**Fig 1A**). We obtained participants' ongoing pain predictions by instructing them to treat the angle as the level of predicted pain regardless of the distance from the initial point, i.e., the center of the rating scale. Prior to heat stimulation, 25 dots appeared on the edge of the semicircular rating scale. Participants were informed that the dots represented other participants' pain ratings from previous experiments about the upcoming stimulus [13,37]. These dots served as pain-predictive cues based on social information. In practice, we displayed 25 dots randomly sampled from a normal distribution for high pain cue (mean: 138.6˚; standard deviation (SD): 9˚) and low pain cue (mean: 39.6˚; SD: 9˚) conditions [8] (**Fig 1B**). There were also trials without cues (i.e., no cue condition). After the cue presentation and jittered inter-stimulus interval, participants experienced 5 different levels of 12.5 s heat stimulation while reporting their continuous pain prediction ratings. The 5 levels of stimulus temperature were fitted for each participant through a calibration procedure prior to the fMRI scans [15] to provide similar levels of pain intensities across participants (see **S1 Fig** for the calibration results and Materials and methods for the detailed experimental procedure). The continuous ratings were collected for 14.5 s, including the 1-s no-heat periods before and after the heat stimulation. After the heat stimulation, we additionally asked participants to rate their overall pain intensity rating for the stimulation.

To assess the effects of the pain-predictive cues and the stimulus intensity on overall pain ratings, we performed a multilevel general linear model (GLM) analysis. The model predicted the overall pain intensity ratings using cue conditions and stimulus intensity as independent variables. We found significant main effects of both the cue level ($\hat{\beta}$ = 0.109, 95% confidence interval (CI) = [0.094, 0.125], $z$ = 3.836, $p < 1.2479 \times 10^{-4}$, bootstrap test with 10,000 iterations) and the stimulus intensity level ($\hat{\beta}$ = 0.106, 95% CI = [0.096, 0.117], $z$ = 3.724, $p < 1.9549 \times 10^{-4}$, bootstrap test) (**Fig 1C**), suggesting that the manipulation of contextual information substantially influenced pain perception [8,13,37].

Next, to investigate the temporal dynamics of the cue and stimulus effects on the continuous ratings, we first resampled the ratings at 50 Hz using linear interpolation. We then

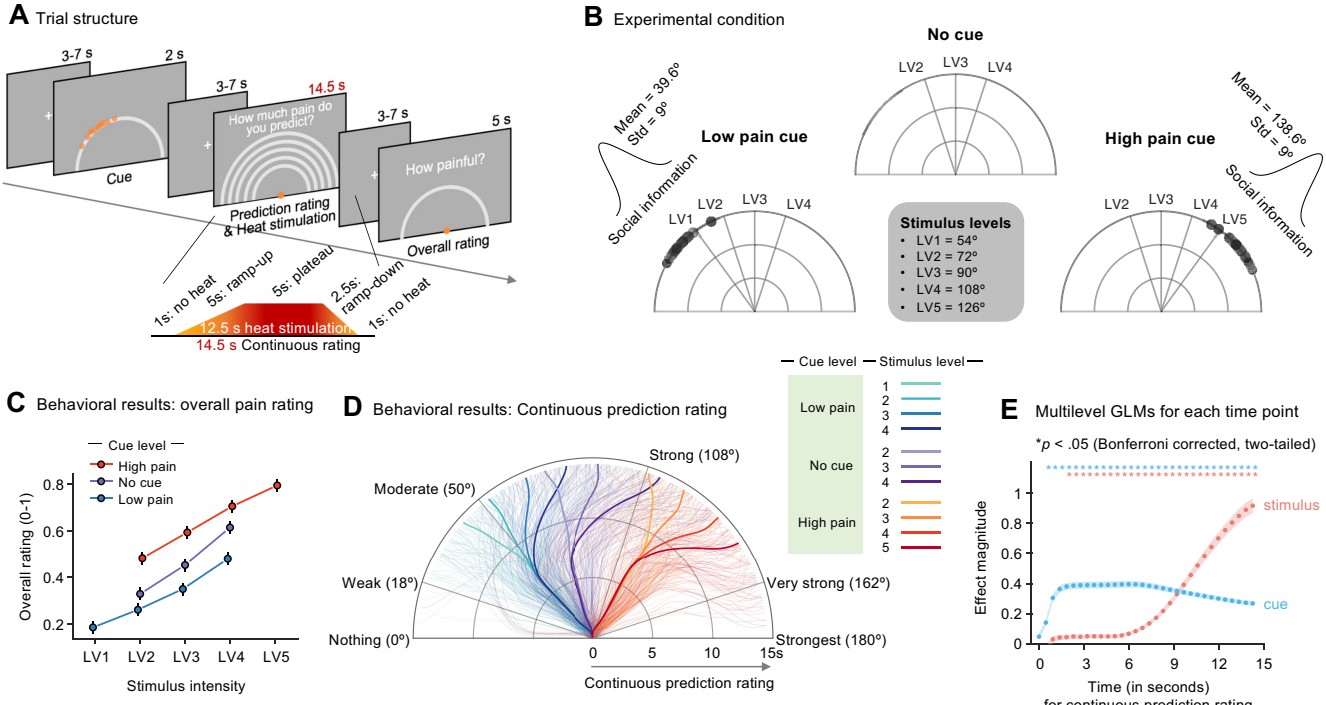

**Fig 1. Experimental paradigm and behavioral results. (A)** Trial structure: Each trial started with a cue phase, except for the "No cue" trials, which began with thermal stimulation. Participants were instructed to continuously report their ratings in response to the following question, "How much pain do you predict?" At the end of each trial, participants provided their overall pain rating. **(B)** Experimental conditions: There were 3 cue conditions: "low pain cue," "no cue," and "high pain cue." Cues consisted of 25 dots randomly sampled from a normal distribution with varying means and an SD. Participants were informed that the dots represented other participants' pain ratings for the upcoming stimulus. We delivered heat stimuli at 5 temperature levels calibrated for each participant to induce similar levels of pain intensity. **(C)** Overall pain ratings: The plot shows the average overall pain ratings as a function of stimulus intensity. Different colors represent different cue conditions. Error bars represent the within-subject standard errors of the mean (SEM) [38]. **(D)** Continuous prediction ratings: Displayed are the trajectories of pain prediction ratings. Thick lines represent the group-average trajectories, while thin lines show the individual-level averages of trajectories. Different colors indicate different experimental conditions. **(E)** Multilevel GLM results: We conducted a series of multilevel GLM analyses, in which we included pain prediction ratings at each time point as a dependent variable and the cue and stimulus intensity conditions and their interaction as independent variables. Given that the continuous ratings were segmented into 32 bins, a total of 32 multilevel GLM analyses were performed. To correct for multiple comparisons, we applied the Bonferroni correction method. Within-subject SEM is denoted by shaded error bands in the plot. Asterisks above the plot indicate time points with significant effects at $p < 0.05$, Bonferroni corrected, two-tailed. The plot uses blue to denote cue effects and red for stimulus effects. The underlying data for Fig 1C and 1E can be found in S1 Data. GLM, general linear model; SD, standard deviation.

averaged the ratings into 32 time bins, ensuring each time bin matched the fMRI sampling rate, TR = 0.46 s. The rating angle, which ranged from 0° to 180°, was transformed to ratings on a scale from 0 to 1. The average trajectory patterns across different conditions indicated that the initial rating direction aligned with the cue location. This direction persisted until roughly 7 s, coinciding with the plateau phase of the heat stimulation (**Figs 1D** and **S2** for each condition). After the plateau, the ratings began to gravitate toward the stimulus intensity level, suggesting that while the cue effects were predominant in the early phase, the stimulus intensity effects became important in the later phase. These observations were corroborated by a series of multilevel GLMs to model the trial-by-trial variations in ratings at each time point based on the levels of cues and stimulus intensity [24,39]. As shown in **Fig 1E**, both the cues and the stimulus intensity significantly influenced the continuous pain prediction ratings throughout the trial, with the exception of stimulus effects being nonsignificant during the initial 1.4 s, which included the 1-s no-heat period (*p*s < 0.05, Bonferroni corrected, two-tailed). Examining the magnitude of the effects, the cue had a larger effect than the stimulus intensity up to 9 s, with its impact gradually decreasing over time after that. Conversely, the influence of

the stimulus intensity increased over time, surpassing the cue effects after around 9 s. In summary, our behavioral analysis showed that the cue and stimulus effects on pain perception exhibited distinct temporal profiles, revealing a gradual shift in their relative importance from contextual to sensory information integration.

## Brain mediators of the cue and stimulus effects on the overall pain ratings

Initially, as done in previous studies [13,15–17], we first sought to identify the brain areas mediating the effects of the cue and stimulus intensity on the overall pain ratings before exploring the detailed temporal dynamics of the integration process. To search for brain mediators, we performed a whole-brain search using multilevel mediation analysis [15,16] (**Fig 2A and 2B**, left). In this analysis, the trial-by-trial experimental condition, such as cue level or stimulus intensity, was included as a predictor ($x$), overall pain rating as an outcome ($y$), and single-trial brain activation maps during the heat stimulation period as mediators ($m$). Brain mediator (or mediation effects) refers to the brain regions mediating the relationship between the effects of stimulus intensity or cue level ($x$) and the overall pain ratings ($y$). When cue

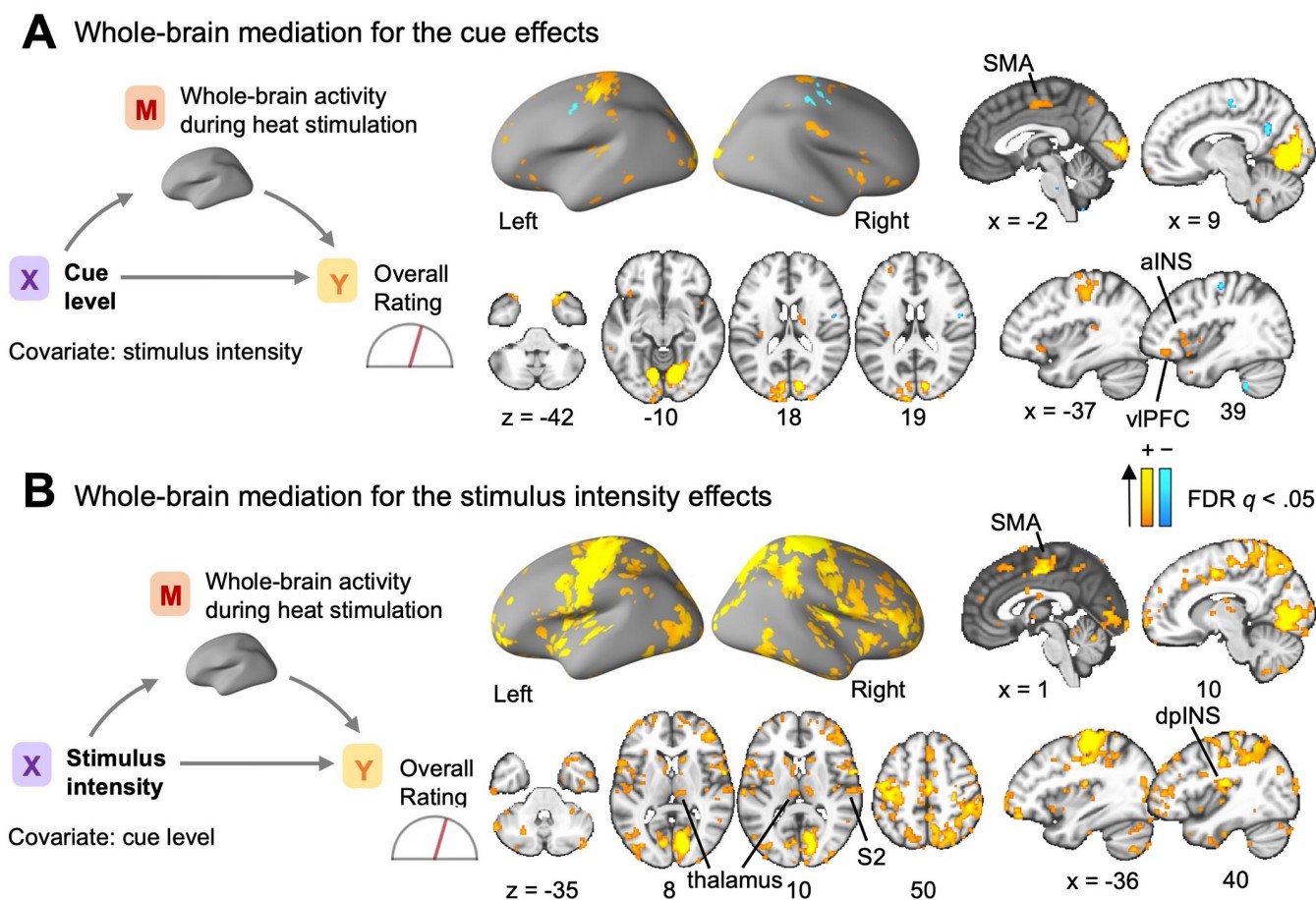

**Fig 2. Brain mediators of cue and stimulus intensity effects on overall pain ratings.** (A) The mediation model assessed how cue levels influenced overall pain ratings, with single-trial brain activation during heat stimulation as the mediator and stimulus intensity as a covariate. The brain map displays significant brain mediators for the cue effects. (B) The mediation model assessed how stimulus intensity influenced overall pain ratings, with brain activation as the mediator and cue level as a covariate. The brain maps were thresholded using an FDR of $q < 0.05$ and a minimum cluster size of $k > 5$ voxels. aINS, anterior insular cortex; dpINS, dorsal posterior insular cortex; FDR, false discovery rate; SMA, supplementary motor area; TP, temporal pole; vlPFC, ventrolateral prefrontal cortex.

condition served as a predictor, stimulus intensity was included as a covariate, and in the model where the stimulus intensity served as a predictor, the cue levels were included as a covariate. The findings showed multiple significant brain mediators for the cue effects at $q < 0.05$, false discovery rate (FDR) corrected, including the aINS, supplementary motor area (SMA), temporal pole (TP), vlPFC, and visual and somatomotor cortex (SMC) areas (**Fig 2A**). Notably, the vlPFC, aINS, and visual cortex regions have been previously associated with cue-induced pain modulation [13,15]. For the effects of stimulus intensity, significant brain mediators at FDR $q < 0.05$ featured some regions overlapping with those identified for cue mediation, including the SMA, SMC, aINS, and visual cortex (**S3 Fig**). However, unique brain mediators distinct from those in cue mediation were also identified, including the dpINS and thalamus. In addition, there was a more pronounced involvement of the SMC areas, including the S2 (**Fig 2B**). These regions have also been previously linked to pain processing in the brain [17,40–43]. Overall, the results of the conventional mediation analysis for the overall pain ratings were largely consistent with findings from previous studies.

## Temporal dynamics of brain mediation for cue and stimulus effects

Finally, to investigate the dynamic integration process of cue and stimulus information in the brain, we utilized temporal mediation analysis, analyzing the temporal dynamics of brain mediation for the cue and stimulus effects on the continuous pain prediction ratings. To this end, we performed the whole-brain multilevel mediation analysis with combinations of 45 TR (= 20.7 s) heat-evoked brain activity from the single-trial finite impulse response (FIR) model and 32 segmented continuous pain prediction ratings (= 14.5 s) (**Fig 3A**; see Materials and methods for details). We included a longer duration for the brain activation maps to accommodate potential hemodynamic response delays. To help interpretation of the results, we segmented the 20.7-s brain activation period into 3 time domains—early (0–6 s), middle (6–13.8 s), and late (13.8–20.7 s). Similarly, the 14.5-s rating period was segmented into the following time domains—cue dominant (1–6 s), transition (6–10 s), and stimulus dominant (10–14.5 s), based on the multilevel GLM results shown in **Fig 1E**. As illustrated in **Fig 3**, these mediation analyses yield high-dimensional outcomes. To manage this complexity, we employed a data-driven approach via independent component analysis (ICA [44]) and derived 5 independent components (ICs) to identify significant and interpretable spatiotemporal domains for brain mediation. Based on the top 2.5 percentile of the temporal weights, we first defined temporal domains and then identified brain mediators for each time domain based on preestablished criteria for selecting brain mediators. Specifically, we selected voxels that survived at FDR $q < 0.05$ with a cluster size greater than 5 voxels and covered at least 5 percent of the defined time domain (see **S4 Fig** for a graphical illustration).

Through the temporal mediation analysis with the cue effects as a predictor, we identified 4 main time domains (as illustrated in **Figs 4A** and **S4B**). These time domains included the mapping between the early and middle phases of heat-evoked brain activity and the cue dominant and transition phases of the continuous pain prediction ratings, suggesting that the early and middle phases of the brain activation are important for mediating the cue effects. The suprathreshold brain regions within these 4 temporal domains included the dorso- and ventro-lateral PFC (dlPFC and vlPFC), intraparietal sulcus (IPS), TP, and visual cortex areas that have been associated with pain predictive cue-induced pain modulation in previous studies [13–15] (**Fig 4B and 4C**). Regarding the large-scale functional networks, the brain regions associated with early mediation of cue effects engaged the dorsal and ventral attention and frontoparietal networks, and their mediation effects waned over time. Conversely, the mediation of cue effects of the somatomotor (particularly in the left hemisphere) and visual networks increased

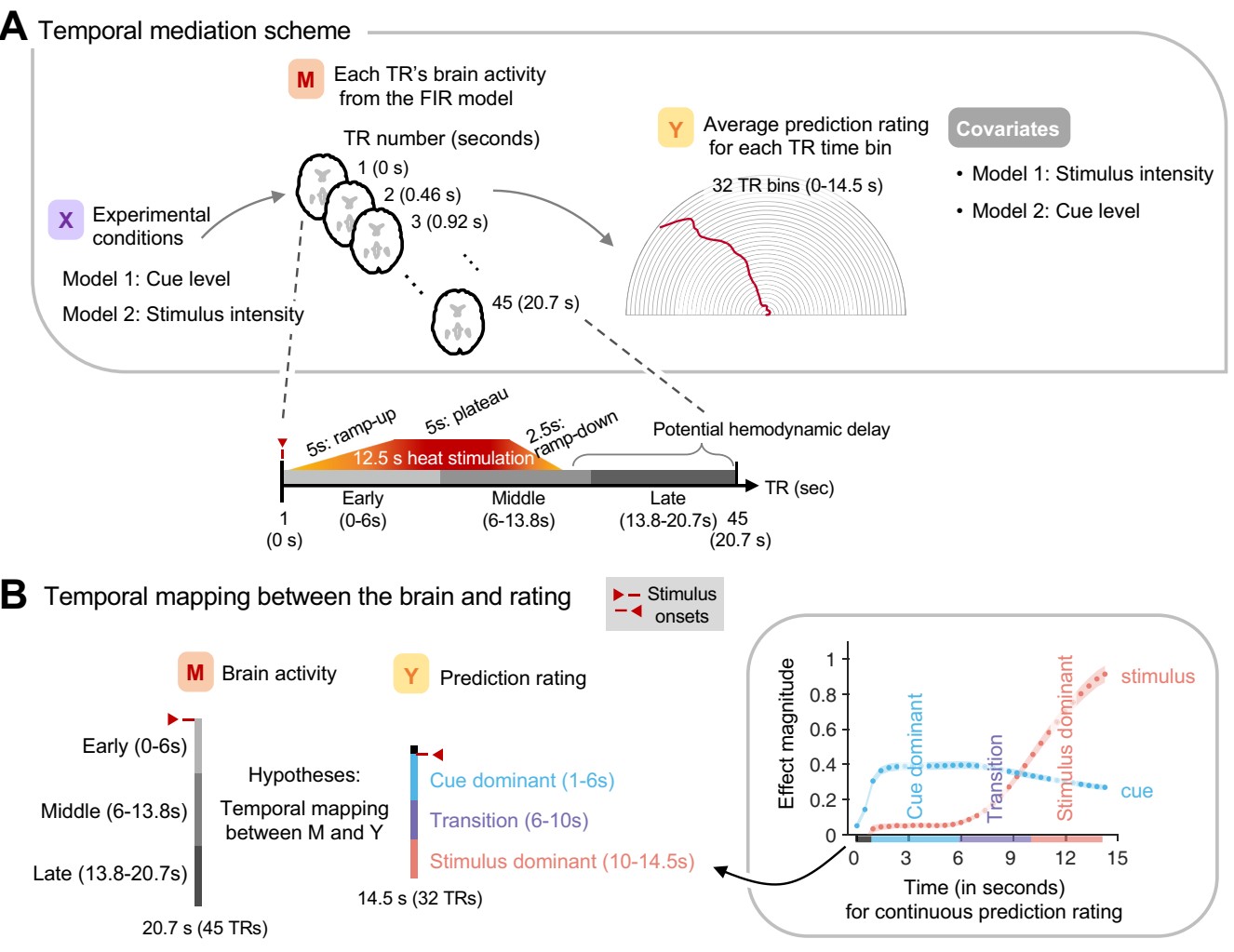

**Fig 3. Temporal mediation analysis. (A)** A schematic representation of whole-brain multilevel mediation analysis with 45 TR heat-evoked brain activation patterns from a single-trial FIR model and 32 time-segmented pain prediction ratings. The temporal composition of the 45 TRs is shown below. The temporal mediation analysis quantified the influence of experimental conditions (*x*) on brain activity (*m*), and how these brain activity patterns relate to continuous pain prediction ratings (*y*) accounting for the experimental conditions at each time bin. In the stimulus intensity model, cue level served as a covariate, and conversely, stimulus intensity was a covariate in the cue mediation model. **(B)** To help interpretation of the temporal mediation results, we divided the brain activity into 3 time domains—early (0–6 s), middle (6–13.8 s), and late (13.8–20.7 s). We also divided the 14.5-s rating period into the following time domains—cue dominant (1–6 s), transition (6–10 s), and stimulus dominant (10–14.5 s) based on the multilevel GLM results. FIR, finite impulse response; GLM, general linear model.

over time. This suggests a transition in the brain systems' engagement from higher-order cognitive processes to visuomotor functions in relation to cue effects.

This idea is further supported by the analysis using the principal gradient map [33] derived from the resting-state fMRI data of the participants in this study (**Fig 4D**). The principal gradient axis, spanning from sensorimotor (unimodal) to association (transmodal) regions, has been proposed as a key functional organization in the human brain [33–36,45]. This axis serves as an intrinsic coordinate system, supporting the progression of information processing from sensory inputs to higher cognitive functions. We used the BrainSpace Toolbox [46] to generate

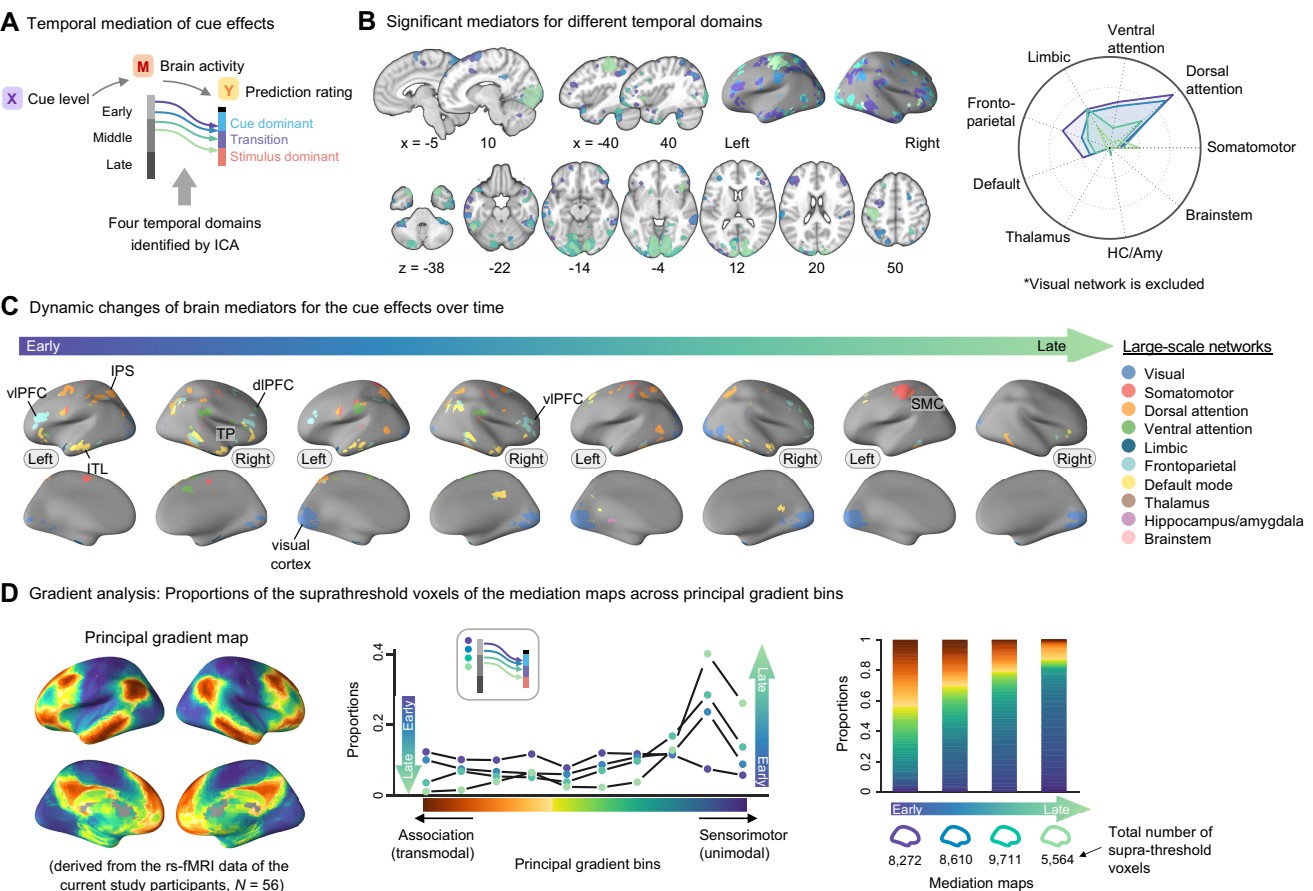

**Fig 4. Temporal mediation analysis results for the cue effects. (A)** A schematic diagram showing 4 temporal domains identified with the ICA (see **S4 Fig**) in the cue mediation model. Lines with different colors represent the mapping between the early and middle phases of heat-evoked brain activity and the cue dominant and transition phases of the continuous pain prediction ratings. **(B)** The brain map displays the supra-threshold brain mediators for the cue effects. The 4 distinct colors in this map correspond to the 4 temporal domains. The radial plot shows the relative proportions of the number of overlapping voxels between the thresholded mediation maps and each network (or region), given the total number of voxels within each network (or region). We did not include the visual network in the radial plot. It is because the visual cortex activation would mainly reflect the task-related features, but it dominated the plot, making it difficult to interpret the activation patterns over time. **(C)** The brain mediation maps corresponding to the 4 temporal domains are color-coded based on large-scale functional and subcortical brain networks. **(D)** (Left) The volumetric functional gradient map was derived from resting-state fMRI data of the current study participants ($N = 56$). This map shows the first connectivity gradient (i.e., principal gradient) represents a major functional axis spanning from association (transmodal) to sensorimotor (unimodal) brain systems (see Materials and methods for more details). (Middle, right) We divided the principal gradient into 2 sets of bins: one set comprising 10 bins using 10-percentile intervals (middle) and another set of 100 bins using 1-percentile intervals (right), creating binary gradient bin masks for each set. We then calculated the proportions of overlap between the thresholded brain maps and these gradient bin masks (see Materials and methods for more details). In the middle panel, the plot shows the overlaps between 4 mediation maps corresponding to 4 temporal domains and 10-bin principal gradient masks. Mediation maps are represented with different dot colors. The gradient transitioning from red to blue represents the principal gradient spectrum. In the right panel, we visualized the overlaps between the same 4 mediation maps and 100-bin principal gradient masks using a stacked bar chart. The underlying data for Fig 4D can be found in S1 Data. HC/Amy, hippocampus and amygdala; dlPFC, dorsolateral prefrontal cortex; fMRI, functional magnetic resonance imaging; ICA, independent component analysis; IPS, intraparietal sulcus; ITL, inferior temporal lobe; SMC, somatomotor cortex; TP, temporal pole; vlPFC, ventrolateral prefrontal cortex.

the principal gradient map based on resting-state data of the same participants (see Materials and methods for more details). We divided the principal gradient into 10 or 100 bins (10-percentile and 1-percentile bins), creating binary masks for each gradient bin. We then calculated the overlap proportions between the supra-thresholded brain maps and these gradient bins [47,48] (see Materials and methods for more details). The results showed that the temporal dynamic patterns in the brain showed a clear transition from association to sensorimotor areas during the mediation of cue effects.

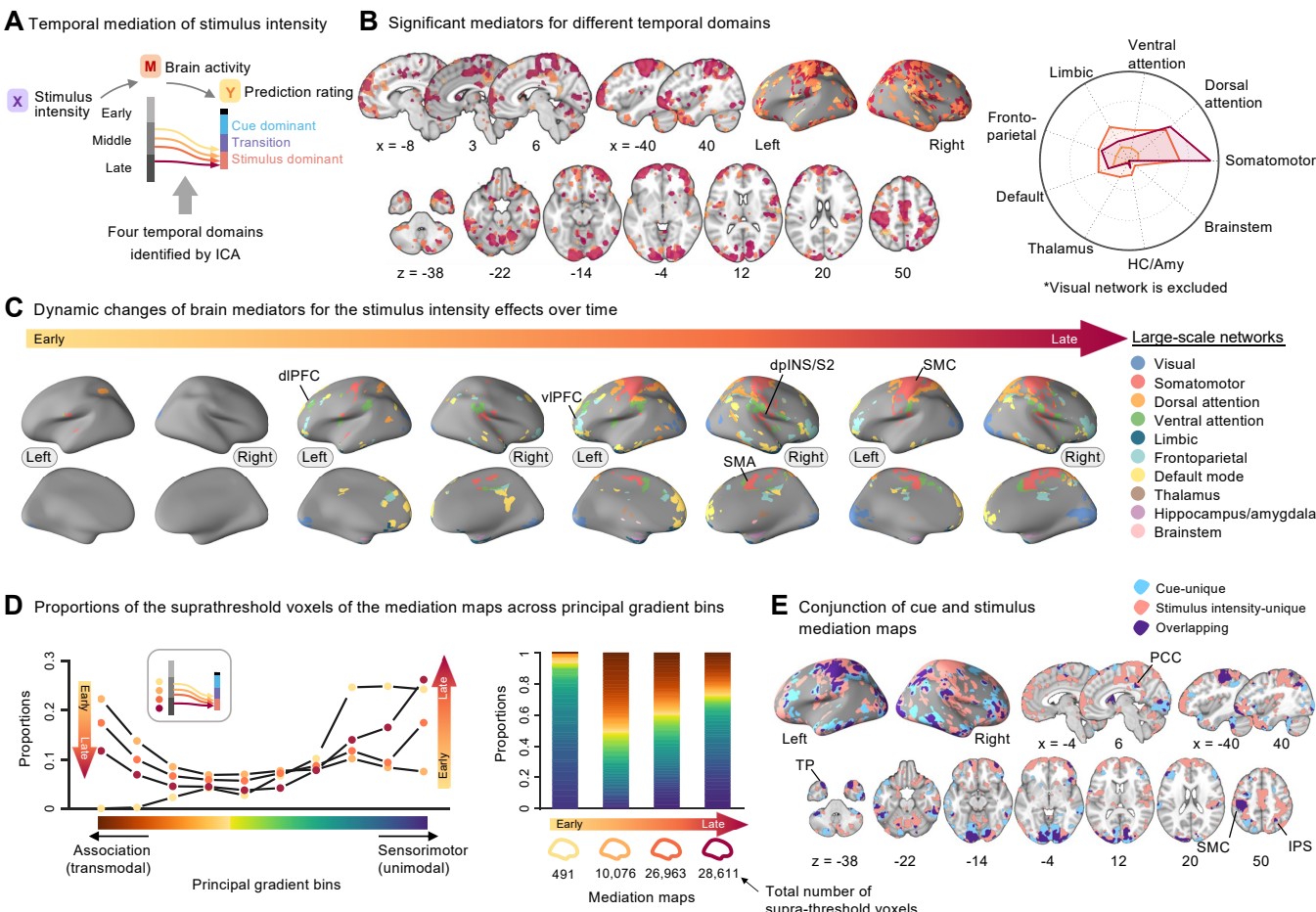

**Fig 5. Temporal mediation analysis results for the stimulus effects. (A)** A schematic diagram showing 4 temporal domains identified with the ICA (see **S4 Fig**) in the stimulus intensity mediation model. Lines with different colors represent the mapping between the middle and late phases of heat-evoked brain activity and the stimulus-dominant phase of the continuous pain prediction ratings. **(B)** The brain map displays the supra-threshold brain mediators for the effects of stimulus intensity. The 4 distinct colors in this map correspond to the 4 temporal domains. The radial plot shows the relative proportions of the number of overlapping voxels between the thresholded mediation maps and each network (or region), given the total number of voxels within each network (or region). We excluded the visual network on radial plot because it reflected task characteristics, making it difficult to understand the patterns over time. **(C)** The brain mediation maps corresponding to the 4 temporal domains are color-coded based on large-scale functional and subcortical brain networks. **(D)** (Left, middle) Similar to **Fig 4D**, we displayed the proportions of overlap between thresholded brain maps and the gradient bin masks. In the left panel, the plot shows the overlaps between 4 mediation maps corresponding to 4 temporal domains and 10-bin principal gradient masks. Mediation maps are represented with different dot colors. The gradient transitioning from red to blue represents the principal gradient spectrum, ranging from association (transmodal) to sensorimotor (unimodal) brain areas. In the middle panel, we visualized the overlaps between the same 4 mediation maps and 100-bin principal gradient masks using a stacked bar chart. **(E)** The conjunction map shows the spatial overlap and unique areas involved in the cue and stimulus mediation. Regions colored in cyan and pink denote brain areas uniquely associated with the mediation of cue and stimulus effects, respectively. The areas where these 2 mediation maps overlap are colored in purple. The underlying data for Fig 5D can be found in S1 Data. HC/Amy, hippocampus and amygdala; dlPFC, dorsolateral prefrontal cortex; dpINS, dorsal posterior insular cortex; ICA, independent component analysis; S2, secondary somatosensory cortex; SMC, somatomotor cortex; vlPFC, ventrolateral prefrontal cortex.

We next identified 4 main time domains and brain mediators of the stimulus effects (**Fig 5A**). These time domains showed the mapping between the middle-to-late phases of brain activity and the stimulus-dominant phase of ratings. The supra-threshold brain regions within these time domains also included the dl/vlPFC, IPS, TP, and visual cortex areas as in the cue mediation maps, but the stimulus mediation maps additionally showed strong brain mediations in the somatomotor network regions, such as the SMA, dpINS/S2, and SMC areas (**Fig 5B and 5C**). Importantly, the total number of significant voxels increased over time,

suggesting that the stimulus mediation effects intensified over time, particularly at the later stages of integration. In terms of the large-scale functional networks, brain regions within the somatomotor and visual networks showed heightened mediation effects in the later phase. In addition, the dorsal attention and frontoparietal network regions also demonstrated increased involvement during the later phase of integration. Additional analyses with the principal gradient map showed a similar pattern to the cue mediation results—i.e., the transition from association to sensorimotor brain areas over time was observed in the stimulus mediation maps (**Fig 5D**).

When we compared the cue and stimulus mediation maps, there were multiple overlapping brain regions (**Fig 5E**). For example, motor and visual cortex regions appeared to mediate both the cue and stimulus effects, which may be related to task-related factors, such as operating a rating apparatus and looking at the screen. This interpretation was further supported by the activation patterns of the simple motor task (**S5 Fig**; for more details on the simple motor task, please see Materials and methods). In addition to these regions, there were other overlapping brain mediators, such as subregions within the vlPFC, IPS, TP, inferior temporal lobe (ITL), and posterior cingulate cortex (PCC), which may play a role in maintaining the cue information throughout the whole integration period, facilitating the integration process of the cue and stimulus information. There were also brain regions that exclusively contributed to the mediation of either the cue or stimulus effects. For example, brain regions within the frontoparietal network, including the dlPFC, were among the unique brain mediators for the cue effects. For the stimulus effects, multiple brain regions within the somatomotor network, such as the dpINS, SMA, and right SMC regions, appeared as unique brain mediators. Notably, the anterior and posterior parts of the right insular cortex differentially mediated the cue and stimulus effects, respectively, which was consistent with the mediation analysis results on the overall pain ratings as shown in **S3 Fig** and previous studies [9,49,50]. To summarize the temporal dynamics of the overlapping brain regions' mediation effects, some selected regions' temporal mediation profiles were visualized in **Fig 6**. The results show that many of the overlapping brain mediators showed significant mediation during the early phase of the cue effects and the late phase of the stimulus effects, suggesting their role in representing prior and posterior information of pain.

## Discussion

The current study investigated the spatiotemporal integration of contextual and sensory information within the cortical hierarchy in pain experience. Our findings provide comprehensive temporal profiles of behavioral and neural processes underlying the complex interplay between cognitive and sensory components of pain [34,51]. The pain experience can be conceptualized as a continuous updating process, in which predictions about the pain state are continuously updated by integrating contextual information with incoming sensory inputs. Particularly in this study, we examined the temporal dynamics of brain mediation underlying this continuous integration process for pain by utilizing novel behavioral and analysis methodologies, including a semicircular pain rating scale and a temporal mediation analysis method.

In this study, we introduced a semicircular rating scale, presenting a new method of conducting trajectory-based analyses on rating behaviors to quantitatively describe and visualize the temporal dynamics of pain perception. A common challenge in utilizing continuous ratings in fMRI studies with the conventional linear rating scale lies in the potential confounding effects arising from the correlation between rating-related motor responses and pain intensity [18]. For example, to indicate higher pain intensity, participants are required to move the cursor further to the right or upwards [19,20]. In the semicircular rating scale, the use of angle

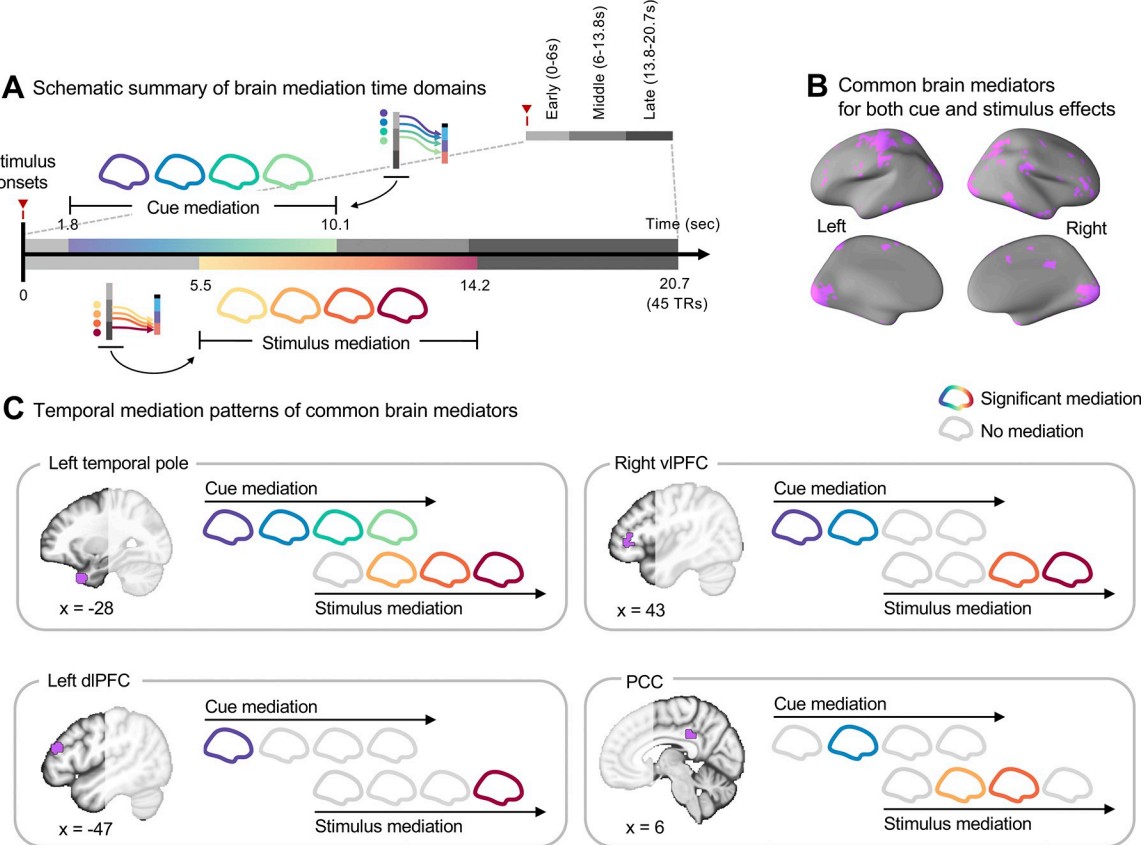

**Fig 6. Temporal mediation patterns of shared brain mediators. (A)** The schematic serves as a visual summary, depicting the brain mediation time domains identified through ICA. The duration of mediation effects—cue mediation from 1.8 to 10.1 s poststimulus onset, and stimulus mediation from 5.5 to 14.2 s—is marked on the timeline based on the ICA results. **(B)** The brain map shows common brain mediators (cluster extent $k > 2$) for both cue and stimulus effects. **(C)** The specific time domains where selected common brain mediators exhibit significant mediation effects. Colored brains correspond to periods of significant mediation, whereas gray indicates nonsignificant mediation. dlPFC, dorsolateral prefrontal cortex; ICA, independent component analysis; PCC, posterior cingulate cortex; vlPFC, ventrolateral prefrontal cortex.

rather than distance to indicate the level of pain allows us to minimize the impact of movement-related confounds. In addition, the trajectory information derived from pain ratings can reveal cognitive states through the quantification of ongoing cognitive processes that would otherwise remain hidden [12,21]. For example, previous studies have demonstrated that the rating trajectory can provide information about latent cognitive dynamics, such as number comparison [22], food choice influenced by health-related cues [24], and face perception influenced by stereotypes [23]. Unlike button press-based assessments that primarily provide reaction time or performance data, trajectory-based measurements can offer more detailed and richer information about the temporal patterns of ongoing cognitive processes, which may vary across contexts and among individuals.

Furthermore, we implemented temporal mediation analysis to more effectively model the ongoing effects of cues and stimuli on the brain and the rating trajectory. This analysis method allows us to go beyond conventional mediation analysis approaches [13,15,16], which typically focus on a snapshot of the brain–behavior relationship. In addition, we adopted a data-driven approach to identify distinct temporal domains important for the cue and stimulus effects on the brain, specifically utilizing ICA [44]. The integration of the temporal mediation methods

with ICA offers a new quantitative way to explore the temporal dynamics of the brain–behavior relationship.

The analysis of the rating trajectories highlighted the gradual changes in the effect magnitudes of both the cue and the stimulus—from the cue-dominant phase through the transition phase to the stimulus-dominant phase. This finding, although could potentially offer an important mechanistic perspective on pain perception, has not been clearly demonstrated in previous research. Prior studies have mainly focused on the overall effects of cue and stimulus intensity on pain ratings, often overlooking how these effects unfold over time [8,13,15,52,53]. The current study, however, views pain perception as a continuous dynamic process [11], in which incoming sensory information continuously updates priors, thereby shaping the pain experience [6,54,55]. An additional analysis correlating continuous pain prediction ratings with overall post-stimulus pain ratings revealed a gradual increase in similarity over time (**S6 Fig**), demonstrating the temporal progression in updating predictions for the dynamic construction of the pain experience.

The results from our temporal mediation analyses indicate that brain regions within the frontoparietal and dorsal attention networks, such as the dl/vlPFC and IPS, were among key brain mediators of the cue effects during the early time domains. This aligns with previous studies indicating the involvement of these networks in sensory integration and the attentional modulation of pain [56–58]. Moreover, they have been identified as mediators in the context of social influences on pain [13]. Further examination of the signs of paths *a* and *b* for the cue mediation effects (**S7A Fig**) revealed that most regions within the frontoparietal and dorsal attention networks showed positive associations with cue levels and pain prediction ratings, highlighting their facilitating role in cue-induced pain modulation [56,57]. For the stimulus intensity effects, brain regions within the somatomotor network, such as the SMA, SMC, and dpINS/S2 areas, appeared to be the key brain mediators. The mediation effects of these regions became stronger over time during the middle-to-late phases. These findings are consistent with earlier research suggesting the significance of these brain regions in representing subjective pain intensity [41,43,59–62] and mediating the effects of stimulus intensity [16,17].

Importantly, there were also overlapping brain mediators between the cue and stimulus effects, including the dl/vlPFC, IPS, TP, ITL, and PCC, as shown in **Fig 5E**. Many of these regions demonstrated significant mediation effects spanning the early-to-late periods of the trial (**Fig 6C**). Prior studies have highlighted the crucial roles of the lateral prefrontal cortices, specifically the dlPFC and vlPFC, in encoding contextual information and filtering out irrelevant details [27–32,54]. In addition, the dl/vlPFC have been implicated in mediating the effects of cue and stimulus intensity on pain [15,17]. Additionally, the dlPFC, in conjunction with the TP, has been suggested to shape the affective tone related to short-term memories of pain experience [63], and distinct temporal profiles of the TP and vlPFC have been linked to working memory processes [64]. Together, these findings suggest that the overlapping brain mediators, particularly dl/vlPFC and TP may play crucial roles in maintaining contextual and sensory information as a central integration hub, assembling evidence over an extended timeframe to construct the subjective experience of pain.

Furthermore, the temporal mediation results of our study can be interpreted within the framework of a cortical hierarchy as proposed by Mesulam [34] and further elaborated by many recent studies [33,35,36]. Our findings suggest a dynamic transition from association (transmodal) to sensorimotor (unimodal) regions for integrating cue and stimulus information during pain experience. This transition, from encoding expectations in the association areas to incorporating sensory inputs in sensorimotor regions, aligns with previous studies on cerebral contributions to pain perception [1,3,53,54]. This does not, however, imply that the information flow or integration process is unidirectional. Instead, the continuous and

reciprocal interactions between association and sensorimotor brain regions are crucial for updating current beliefs [10], adapting to new stimuli, such as through habituation, sensitization, and normalization [65], and constructing the subjective experience of pain. However, with the current experimental design and analysis approach, we could not directly test the ongoing bidirectional interactions across the cortical hierarchy, highlighting the need for future studies.

The temporal mediation analyses in our study, while informative, are not without limitations. First, hyperparameters, such as the number of independent components and thresholds for temporal domains and spatial maps, were somewhat arbitrary. To assess the robustness of our results against these arbitrary hyperparameter choices, we conducted a supplementary analysis by varying the hyperparameters and evaluated the spatial similarity with our main brain mediation maps. As shown in **S8 Fig**, the supplementary analysis results yielded brain mediation patterns largely consistent with the main findings. In addition, to account for the possibility of missing important mediators due to the arbitrary hyperparameter choices, we conducted another supplementary analysis using 17 predefined ROIs (**S9A Fig** and Materials and methods), which are known to be associated with pain processing [43,66], self-regulatory strategies [16,67], and pain expectations manipulated by social information [13]. The results suggested that many brain regions showed similar temporal mediation patterns as those identified in whole-brain temporal mediation analysis, including the vlPFC, vmPFC, OFC, and IPS as mediators of cue effects on pain, and the cerebellum, pgACC, SMG, thalamus, dpINS, vlPFC, and IPS as mediators of stimulus intensity effects on pain (**S9B and S9C Fig**). However, it also revealed unique temporal mediation patterns, as described in **S9 Fig**. To expand the current findings and explore additional spatiotemporal patterns of temporal domains, future studies could incorporate a greater number of components and higher spatial resolutions.

The continuous pain prediction task also has some limitations. Most importantly, it was inherently susceptible to head and arm movements due to the motion required for reporting their pain prediction and pain ratings using the MR-compatible joystick. To mitigate the influence of movements on heat-related BOLD signals, we implemented 2 strategies. First, we increased the sample size of the fMRI experiment to 59 participants., exceeding previous fMRI studies on similar topics, which included 38 participants [13], 30 participants (Study 1 of [10]), and 34 participants (Study 2 of [10]). It allows us to reduce the possibility that the current results might be influenced by a small number of outliers with extreme head and/or arm movements. Second, we excluded the trials with 2.5 variance inflation factors (VIFs) as high VIFs indicate potential multicollinearity between head movement and effects of interest. Moreover, excluding high VIFs (>2.5) can minimize the effects of non-interest such as signals of cerebrospinal fluid and white matter, leading to more reliable results. The average number of excluded trials was 3.51 (about 3%; $SD = 2.09$) of the total 108 trials per participant. Thus, we believe our results are unlikely to be substantially affected by movement artifacts. However, future studies would benefit from methods that can alleviate the impact of non-BOLD artifacts, such as multi-echo fMRI [68,69]. Lastly, the semicircular rating scale could introduce some bias as the scale has an unequal rating distance at the center versus the periphery. This leads to small initial movements near the center resulting in large changes in ratings, and large later movements in the periphery resulting in small changes in ratings. To mitigate this, we provided extensive training and clear instructions to participants prior to the experiment (see **S10 Fig** and Materials and methods). In particular, we emphasized that it was the angle, not the distance, that reflected their ratings and that they were not required to reach the outer end of the semicircles. However, future studies should examine the same question using different types of rating scales.

Overall, the current study provides new insights into the spatiotemporal dynamics of pain perception and enhances our comprehension of underlying neural mechanisms. Utilizing

novel tools and analysis approaches, our results support the notion that pain perception is a dynamic process of continuously integrating contextual information with sensory inputs. This underscores the importance of investigating the temporal dimension of pain perception. We hope that our findings will contribute to a more profound understanding of the brain mechanisms responsible for shaping and modulating the experience of pain.

## Materials and methods

### Participants

We recruited a total of 84 healthy participants who were right-handed, had no neurological disorders, and agreed to participate in a pain experiment. Among them, 59 participants (Korean native speakers; $n_{female}$ = 26; mean$_{age}$ = 22.11; range$_{age}$ = 18–28; SD$_{age}$ = 2.44) completed the experiment. Specifically, 84 participants came to the first session that included the pain calibration task [8,15] and a battery of individual difference questionnaires (e.g., demographic, psychological status, and health questionnaires). Thirteen did not continue the experiment because their pain calibration results exceeded the exclusion criteria and 8 participants notified a withdrawal from the next session due to their own personal reasons. Thus, the remaining 63 participants underwent the fMRI task. Among them, 3 participants requested to quit the experiment during the scan, and 1 participant showed pain ratings substantially different from the pain calibration results. Participants were recruited through the university websites and flyers posted on the university buildings. All participants provided signed informed consent prior to the experiment in both sessions and received monetary compensation for their participation. The experiment was approved by the institutional review board of Sungkyunkwan University (IRB number: 2017-05-001) and was performed in accordance with the Declaration of Helsinki. The present study collected data from spring to autumn 2018.

### Thermal stimulation

Thermal stimulation that ranged from 40 to 49.2˚C (baseline: 32˚C) was delivered to a left forearm using a $16 \times 16$ mm ATS thermode (Medoc, Israel). The thermal stimulation had a duration of 12.5 s, comprising a 5-s ramp-up, a 5-s plateau, and a 2.5-s ramp-down (**Fig 1A**). We opted for a slow ramp-up, as a faster ramp-up (e.g., 1 or 2 s) would likely hinder the ability to capture the relevant neural and behavioral features associated with the integration process. During the pain calibration task, we changed the stimulation sites on the forearm for each trial, and similarly, for each experimental run during the fMRI task. Four skin sites were used for the former and 3 for the latter. A single highest temperature of heat stimulation was delivered before the pain calibration task (e.g., 49.2˚C) and each run during the fMRI experiment (e.g., temperature corresponding to the stimulus intensity level 5 from the result of the pain calibration task) to avoid the initial habituation of the skin site to contact heat [65] and to confirm the working of heat stimulation delivery.

### Semicircular rating scale

To collect continuous pain prediction ratings, we devised a new rating scale, named the semicircular rating scale. For continuous ratings, we made the scale surrounded by multiple semicircles to imply that it is a continuous field, while a single semicircle was used for the overall rating (**S11 Fig**). An important feature of the semicircular rating scale is that the starting point is equidistant from all possible ratings, addressing an issue of intensity rating being confounded with the cursor movement distance. An orange-colored dot was located at the center of the scale, and the dot can be operated by input devices such as a joystick, which we used in

this study. Participants were asked to report their ratings using the angle from the left segment of the semicircle base by manipulating the orange-colored dot (see **S11 Fig**). The rating trajectory was recorded by both the *x* and *y* coordinates of the dot and the angle at a sampling rate of 60 Hz. To define the anchors of the scale, we modified the generalized labeled magnitude scale (gLMS [70]) to make participants use the entire space of the scale. The anchors consist of no sensation (0˚), weak (18˚), moderate (50˚), strong (108˚), very strong (162˚), and strongest imaginable sensation (180˚; see **Figs 1D** and **S11**).

## Experimental procedure

The entire experiment unfolded over 2-day sessions. In the first day session, we executed the pain calibration task to induce similar levels of pain experience across participants [8,15]. During the second day session, we conducted the fMRI pain experiment. At the beginning of each session, we provided an overview including the session's duration, the tasks, and the use of the scale. After the overview, we ensured participants understood how to use the rating scale and the upcoming tasks by asking them to explain the scale and the task procedures (see **S10 Fig** for more information). If participants were unable to accurately describe these elements, we provided further explanation until they demonstrated a clear understanding. This aimed to ensure consistent and accurate use of the scale throughout the experiment. We utilized Psychtoolbox (http://www.psychtoolbox.org) and MATLAB (MathWorks) to present stimulus, record ratings, and administer heat stimulation. A more detailed experimental procedure is described in each task section below.

**Day 1: Pain calibration task.** Upon arriving at the laboratory, participants were informed about the procedures of the entire experiment and provided written informed consent. After completing a battery of individual difference measures—including demographic information, behavioral tendency, emotional states, and traits (not analyzed in this study)—we conducted the pain calibration task. The task had 3 objectives: (1) to familiarize participants with the heat stimulation and rating procedure, e.g., the use of a joystick and rating scales; (2) to match the levels of subjective pain experience across individuals; and (3) to exclude individuals with either excessively low or high sensitivity to heat stimulation.

In the pain calibration task, participants reported pain intensity ratings for 12 heat stimulations. Initially, 3 predetermined heat temperatures (43.4, 45.4, and 47.4˚C) were delivered in random order. Based on the pain ratings for these stimuli, we fitted a linear regression model to predict pain ratings (dependent variable) using temperature levels (independent variable). For subsequent trials, the temperature levels were determined using this model [8,15]. Specifically, we estimated 3 temperatures corresponding to low, medium, and high stimulus intensity levels based on the linear regression model. Low, medium, and high stimulus intensity levels were operationalized as 30% (i.e., 54˚), 50% (90˚), and 70% (126˚) of the semicircle rating scale. One of these 3 stimulus intensity levels was delivered for the next trial in a pseudo-random order, ensuring no consecutive repetition of the same stimulus intensity. This procedure was repeated for each trial up to the 12th trial—i.e., for each N-th trial, the temperature was determined by fitting a new linear regression model using the temperature intensities and corresponding pain ratings from the first to the (N-1)-th trial. Finally, we determined 5 temperature levels for each participant, corresponding to 5 stimulus intensity levels (30, 40, 50, 60, and 70% of the scale, equating 54˚, 72˚, 90˚, 108˚, and 126˚, respectively; see **Fig 1B**). These 5 temperatures, derived from the final linear regression model of each participant, were used in the fMRI pain prediction task.

If the estimated 5 temperatures fell outside the predefined temperature range, which spanned from 40 to 49.2˚C, or if the $R^2$ of the final linear regression model was lower than 0.4,

we excluded the participants from further experimentation. In addition, based on the $R^2$ for each skin site, we selected the top 3 skin sites, which were used in the subsequent fMRI experiment. **S1 Fig** shows the pain calibration task results of the remaining participants. For the calibration task, participants used a joystick to report pain ratings, which helped them become familiar with the setting for the fMRI experiment. The lighting and temperature of the room in which the pain calibration task was conducted were maintained similarly to those in the MRI room. To administer heat stimulations to the same skin sites in the subsequent fMRI task, we photographed the left forearm of each participant after completing the pain calibration task.

**Day 2: fMRI experiment.** On day 2, we conducted an fMRI experiment, which included 1 resting-state run, 2 simple motor task runs, 6 pain prediction task runs, and a structural scan (T1). First, we acquired structural brain images. During the structural scan, we also administered a pain prediction task with 8 trials to help participants become familiar with the task. In addition, we provided cues consistent with the subsequent stimulus intensity to enhance the cue-induced expectation effects. During this practice, heat stimulations were delivered to a skin site that was not selected from the pain calibration task. Second, we conducted a resting-state run (1 run), lasting approximately 6 min and 10 s (810 TRs). Participants were instructed to gaze at a fixation cross on a dark gray screen. Third, we administered a simple motor task (2 runs), wherein participants were instructed to move the orange-colored dot to a specific location on a semicircular rating scale using an MR-compatible joystick. The simple motor task runs were positioned prior to the first and fourth pain prediction task runs.

Fourth, we conducted the continuous pain prediction task (6 runs). Each run comprised 18 trials (i.e., a total of 108 trials), and each trial included a series of the following events: (1) pain-predictive cues; (2) pain prediction rating and heat stimulation; and (3) overall pain rating. More specifically, during the cue event, 25 dots on the semicircular rating scale were displayed prior to heat stimulation. Participants were informed that the dots represented other participants' pain ratings for the upcoming stimulus. Prior studies showed that no additional learning procedure was needed to learn these social information-based cues, thereby simplifying our experimental procedure [13,37]. In practice, we generated 25 dots for each condition by sampling from a normal distribution with different means and a shared SD (mean = 39.6° and 138.6° corresponding to "low pain cue" and "high pain cue," respectively, with an SD = 9°). Some trials proceeded without cues (the "no cue" condition; see **Fig 1B**).

During the pain prediction rating and heat stimulation event, a rating scale with multiple semicircles and an orange-colored dot at the scale's center was displayed 1 s before the delivery of the heat stimulus. Participants were instructed to continuously report their ratings in response to the following question, "How much pain do you predict?" We informed participants that they were not required to reach the outer end of the semicircles and that the angle, not the distance from the starting point, reflected their current pain prediction. In addition, we instructed participants to start reporting their pain prediction as soon as the trial began, ensuring they provided their ratings throughout the entire trial. We imposed a limit on the speed of the dot movement to ensure participants did not reach the outer end of the semicircles before the heat stimulus reached its plateau.

During the overall pain rating event, participants were instructed to report their ratings in response to the following questions, "How painful was it?" (self-pain question) or "How painful would it be for others?" (other-pain question) within 5 s. Even if the participant reported pain before the end of the 5 s, the blank screen remained until the end of the 5 s. Within a run of 18 trials, the self-pain question was asked for 11 trials and the other-pain question for 7 trials. The other-pain question was included to address a separate research question that is not covered in the current manuscript. The 11 trials with the self-pain question included all combinations of stimulus intensity levels and cue conditions, allowing us to analyze the cue effects

on the overall pain ratings (**Figs 1C** and **2**). A fixation cross was displayed during the jittered inter-trial and inter-stimulus intervals, which lasted between 3 and 7 s, totaling 15 s for each trial (see **Fig 1A**). The long and variable intervals were designed to effectively dissociate the BOLD signals associated with the cue and the noxious stimulus [71] and to reduce participants' ability to predict the timing of cue presentation and stimulus delivery, thereby minimizing habituation. The trial sequence was pseudo-randomized to prevent the same experimental condition from occurring consecutively.

## fMRI data acquisition and preprocessing

Both functional and structural images were obtained using a Siemens 3.0 Tesla Magnetom Prisma at the Center for Neuroscience Imaging Research in Sungkyunkwan University utilizing a 64-channel head coil. High-resolution anatomical T1-weighted images were acquired using the MPRAGE protocol (TR = 2,400 ms; TE = 2.34 ms; flip angle = 8; field of view = 224 mm; voxel size = 0.7 mm). We acquired functional images using the T2*-weighted echo-planar image (EPI) protocol (TR = 460 ms; TE = 27.20 ms; multi-band factor = 8; field of view = 220 mm; flip angle = 44˚; the number of interleaved slices = 56; voxel size = 2.7 mm$^3$; pre-scan normalize; iPAT = none; anterior-to-posterior phase encoding direction). For all functional scans, we discarded the initial 18 TR images to allow for image intensity stabilization (see **S12 Fig** for the temporal signal-to-noise ratio map). Prior to statistical analyses, we implemented the following preprocessing steps: outlier detection, realignment, distortion correction, coregistration and normalization, and spatial smoothing. (1) Outlier detection: outliers were identified in each image based on Mahalanobis distances and the root mean square of successive differences to eliminate intermittent gradient and severe motion-related artifacts. These types of artifacts are present to some degree in all fMRI data. For Mahalanobis distance-based outlier detection, we computed the distances for the matrix of concatenated slice-wise means and SDs over time and then identified images that exceeded 10 mean absolute deviations based on moving averages with a full width at half maximum (FWHM) of 20 image kernels as outliers. Using the root mean square of successive differences across volumes, images exceeding 3 SDs from the global mean were identified as outliers. Time points flagged as outliers by either detection method were included as nuisance covariates. (2) Realignment: functional images were realigned to the first single-band reference (SBRef) images, and 6 movement parameters (*x*, *y*, *z*, *pitch*, *roll*, and *yaw*) were estimated for each run. (3) Distortion correction: susceptibility-induced image distortion was corrected using TOPUP tool in the FMRIB Software Library (FSL). (4) Coregistration and normalization: we first coregistered the T1 structural images to the first SBRef functional image, segmented and normalized the structural images to the standard brain template, which was the MNI space, and then normalized the functional images to the MNI space using the parameters obtained from the T1 normalization with the interpolation to 2 mm$^3$ voxels. (5) Smoothing: we smoothed the normalized functional images with a 5-mm full-width-at-half-maximum Gaussian kernel. Except for the distortion correction, the functional and structural data were preprocessed using Statistical Parametric Mapping (SPM12; Wellcome Trust Centre for Neuroimaging). The fMRI data of 1 participant was excluded due to an unexpected technical error. Thus, the remaining 58 participants' data were further analyzed.

## Continuous rating data preprocessing

The sampling rate of the rating trajectory was 60 Hz, which was the refresh rate of a display projector (Propixx, VPixx Technologies; resolution of 1,920 × 1,080 pixels). However, due to occasional instability in the refresh rate, the number of data points was not identical across

trials. Thus, we resampled the ratings to 50 Hz using linear interpolations [24], yielding 726 data points for the pain prediction ratings, and applied the Gaussian smoothing (window size = 71) to filter out the irregularities such as noise or jitter of the rating trajectory. We then segmented the continuous pain prediction ratings into 32 time-bins and averaged the ratings for each bin. This resulted in each time bin being 460 ms, aligning with the temporal resolution of our fMRI data. Lastly, we normalized the angle, which ranged from 0˚ to 180˚, to a scale from 0 to 1. For the overall pain ratings, we also resampled the ratings to 50 Hz using linear interpolations and averaged the rating trajectories at the last 1 s rating as the overall pain ratings.

## Multilevel general linear model (GLM) analysis

To evaluate the effects of the pain-predictive cues and the stimulus intensity on ratings, we performed multilevel GLM analyses based on the trial-level behavioral data. In the GLM predicting the overall pain ratings, the cue conditions were coded as −1, 0, and +1, corresponding to low pain cue, no cue, and high pain cue, respectively. The 5 levels of stimulus intensity were coded as 1, 2, 3, 4, and 5. Additionally, an interaction term between the cue and stimulus intensity was included in the model. We used the averaged ratings over the last 1 s of the overall pain rating trajectory for the self-pain question trials. For significance testing, we conducted bootstrap tests with 10,000 iterations.

In the GLM predicting the continuous pain prediction ratings, we used ratings from each time point as an outcome variable. Considering that the continuous ratings comprised 32 time bins, a total of 32 multilevel GLM analyses were conducted. In this analysis, to help comparisons of the effect magnitudes between the cue and stimulus intensity effects, we coded the cue conditions and stimulus intensity levels based on the normalized rating scale (i.e., 0–1 for 0˚–180˚). More specifically, the low pain cue, high pain cue, and no cue conditions were coded as 0.22 (39.6˚), 0,77 (138.6˚), and 0, respectively, and the stimulus intensity levels 1 to 5 were coded as 0.3 (54˚), 0.4 (72˚), 0.5 (90˚), 0.6 (108˚), and 0.7 (126˚), respectively. These coding corresponds to the positions of the stimulus intensity levels and the mean positions of the cues on the semicircular rating scale (**Fig 1B**). This coding scheme allows us to capture cue and stimulus effects in the same unit, making the beta coefficients comparable. For significance testing, we conducted bootstrap tests with 10,000 iterations and corrected the results for multiple comparisons using the Bonferroni correction. In addition, we compared the current results (**Fig 1E**) with those obtained using a different coding scheme of −1, 0, 1 for low, no, and high pain cues, and 1, 2, 3, 4, and 5 for the stimulus levels. Although the absolute magnitudes of the beta coefficients were influenced by the coding schemes, the correlation between the beta coefficients was nearly perfect ($r$s > 0.99), indicating that the use of different coding schemes did not affect the statistical outcomes.

## fMRI single-trial analysis

We estimated single-trial heat-evoked brain activity using a GLM design matrix with separate regressors for each trial, as in the "beta series" approach [72]. In this analysis, we modeled the heat stimulation epoch of each trial with a boxcar function regressor, convolved with a canonical hemodynamic response function, and included the regressors for all trials in a single design matrix. We also included additional event regressors, including cue presentation and overall pain rating, and the nuisance covariates, including dummy coding regressors for each run (i.e., run intercept), linear drift across time within each run, indicator vectors for outlier time points, 24 head motion parameters, including 6 movement parameters (*x*, *y*, *z*, *pitch*, *roll*, and *yaw*), their mean-centered squares, their derivatives, and squared derivatives, and 5 principal

components of cerebrospinal fluid and white matter signal. In addition, we applied a high-pass filter with a cutoff of 180 s. The single-trial brain activation maps derived from this analysis were included as mediators in whole-brain multilevel mediation analysis (**Fig 2**).

As single-trial estimates could be affected by acquisition artifacts occurring during that trial (for example, sudden motion, scanner pulse artifacts, etc.), we calculated trial-by-trial VIFs (a measure of design-induced uncertainty due to collinearity with nuisance regressors) using the design matrix. Any trials with VIFs that exceeded 2.5 were excluded from subsequent analyses [10,13,16]. The average number of excluded trials was 3.51 (approximately 3%) with an SD of 2.90.

## fMRI single-trial finite impulse response (FIR) analysis

We also conducted the single-trial FIR analysis. Instead of using a boxcar regressor for each trial, we employed a stick function regressor for each TR. We modeled a total of 45 TRs for each trial, which equated to 20.7 s (= 45 TRs × 0.46 s) from the onset of the heat stimulation. Other nuisance covariates remained consistent with the fMRI single-trial model described above. TR-level brain activation maps were included as a mediator in the multilevel temporal mediation analysis (**Fig 3A**).

## Whole-brain multilevel mediation analysis

To search for the brain regions mediating the effects of pain-predictive cues or stimulus intensity on the overall pain ratings, we conducted a whole-brain search for significant mediators with the multilevel mediation models. We tested the same 2 models described in the previous section with each voxel's brain activity as a mediator. We conducted bootstrap tests with 10,000 iterations for significance testing and corrected the results for multiple comparisons using an FDR $q < 0.05$ [73]. For more details of the multilevel mediation analysis, please see refs. [15,16,74].

## Whole-brain multilevel temporal mediation analysis

We performed a whole-brain multilevel temporal mediation analysis to investigate the temporal dynamics of brain mediation for the cue and stimulus effects on the continuous pain prediction ratings. In this mediation model, Path *a* represents the effects of experimental conditions (*x*) on heat-evoked brain activity (*m*) at each time point. Path *b* is the effects of brain activity (*m*) on continuous pain prediction ratings (*y*) for a specific TR time bin above and beyond the effects of experimental conditions. Path *a*×*b* is the mediation effects, defined by the product of Paths *a* and *b*. Testing the significance of Path *a*×*b* is equivalent to a statistical test of Path *c* minus Path *c'*, where Path *c* indicates the effects of experimental conditions (*x*) on pain prediction ratings (*y*), and Path *c'* represents the effects of experimental conditions (*x*) on pain prediction ratings (*y*) after controlling for the effects of brain activity (*m*) on pain prediction ratings (*y*).

The temporal mediation analysis entails testing multilevel mediation models for all combinations of 45 TR (= 20.7 s) heat-evoked brain activity derived from the single-trial FIR model and 32 segmented continuous pain prediction ratings (= 14.5 s). We conducted the analysis for each combination separately and then combined the results, creating 45 × 32 result matrices (see **S4A Fig**). Brain data were resampled to a resolution of 4 mm³ using FLIRT in FSL to reduce computational load, resulting in 24,860 voxels for the whole brain. We subsequently applied temporal smoothing to the resampled brain data using a Gaussian kernel with a window size of 5 via the smoothdata.m function in MATLAB (MathWorks), to reduce noise at the TR level. We eliminated uninterpretable time domains from further analyses, including the

mediation effect of later brain activity on earlier ratings or the influence of stimulus intensity on ratings preceding thermal stimulation. These excluded time domains appear as blank in the result matrices depicted in **S4A Fig**.

## Independent component analysis

To identify temporal domains for brain mediation, we employed ICA. In general, the ICA allows us to decompose complex patterns of data into statistically independent components. Here, ICA was used to identify independent temporal domains and corresponding brain mediators, revealing the mediation effects of certain brain regions on pain prediction ratings at a specific time domain. The temporal mediation result matrices, specifically $p$-values for Path $a \times b$, served as the input for the ICA. To make the input matrix, we first vectorized the mediation result matrices (i.e., $p$-values for Path $a \times b$) for each voxel and converted them into $-\log_{10}(p)$. Next, we concatenated these vectors across all voxels, thereby aggregating the data from the whole brain. We then obtained temporal and spatial component weights using the GIFToolbox with the fastICA algorithm (# of components = 5).

The analysis yielded 2 sets of 5 components, one for spatial weights (24,860 [# of voxels] × 5 [# of components]) and the other for temporal weights (880 or 602 [# of temporal mapping] × 5 [# of components]) (see **S4A Fig**). To identify the brain mediators and relevant temporal domains for each component, we first thresholded the temporal weights with the top 2.5 percentile. The resulting 5 thresholded temporal weights could be described using a river plot connecting the brain activity and pain ratings (**S4A Fig**). We then identified brain mediators related to these temporal domains based on the following 3 criteria: (1) voxels that survived the FDR correction for multiple comparisons at $q < 0.05$; (2) regions with at least 5 contiguous voxels; and (3) the survived voxels should cover at least 5% of the defined temporal domain. The resulting thresholded ICA component weights can be found in **S4B and S4C Fig**. One of the identified components was excluded from further analysis due to its strong association with the visual network, potentially reflecting task-related processes. For a visualization purpose, brain mediators were resampled at 2 mm$^3$ and spatially smoothed with a 2-mm full-width-at-half-maximum Gaussian kernel.

## Principal gradient analysis

To further interpret the brain mediation results based on a cortical hierarchy, we first derived the functional gradient map (only the first gradient) using a resting-state fMRI from the current study participants ($N = 56$). The gradient map was constructed using BrainSpace Toolbox [46] (see below for more details and **S13 Fig**). We estimated the principal gradient using the method employed by ref. [33]. We divided the principal gradient into 10 or 100 bins (using 10-percentile and 1-percentile bins, respectively), resulting in 10 or 100 binary gradient bin masks. Next, we calculated the overlapping proportions between the supra-thresholded brain maps and the gradient bins.

## Building a volumetric gradient map using a resting-state fMRI data

Our study opted for a volume-based fMRI analysis approach, contrasting with the surface-based methods used in the original functional gradient study [33]. This discrepancy made it difficult to directly apply the publicly available functional gradient map to our data. To circumvent this problem, we created our own volumetric gradient map using resting-state fMRI data from the participants of the current study ($N = 56$) using the BrainSpace toolbox [46]. From our original 59 participants, we had to exclude 3 participants from this analysis due to technical issues, such as preprocessing errors, resulting in $N = 56$. To create the volumetric gradient

map, we first resampled the resting-state fMRI data to a voxel size of 3-mm$^3$ to reduce the computational load to a manageable level. We then computed the functional connectivity based on the resting-state fMRI activity with the gray matter mask. Finally, the functional connectivity matrices from 56 participants were averaged, resulting in a single averaged connectivity matrix with a size of 59,026 × 59,026 (voxel-by-voxel). This averaged connectivity matrix was then submitted to the BrainSpace toolbox [46] with the same parameters as the original study [33] (i.e., dimension reduction technique: diffusion embedding, kernel: normalized angle, sparsity: 0.9; see **S13 Fig**).

## Large-scale functional network overlap analysis

The radial plots in **Figs 4** and **5** show the relative proportions of the number of overlapping voxels between the thresholded mediation maps and each network (or region) given the total number of voxels within each network (or region). We used the Buckner group's parcellations to define large-scale functional brain networks, including 7 networks within the cerebral cortex [75], cerebellum [76], and basal ganglia [77]. We also added the thalamus, hippocampus, and amygdala from the SPM anatomy toolbox and the brainstem region [78], as shown in **S14 Fig**.

## Temporal mediation analysis with varying parameters

To assess the robustness of the temporal mediation analysis results presented in **Figs 4** and **5**, we performed a supplementary analysis with different hyperparameters and evaluated the spatial similarity with the original mediation maps. Specifically, we tested: (1) the number of ICA components = 15 and 30 (the original parameter was 5); (2) the threshold of temporal weights = 1% and 5% (original parameter = 2.5%); and (3) the time coverage of the suprathreshold voxels = 2.5% and 7.5% (original parameter = 5%), while keeping all other analysis parameters consistent with **Figs 4** and **5**. Then, we calculated spatial similarities between brain mediation maps, including results from our main findings in **Figs 4** and **5** and new analysis results using Dice coefficients. These similarities were visualized in 2D spaces via multidimensional scaling (MDS), which represents the distance between brain mediators. See **S8 Fig** for results.

## Simple motor task

Participants completed a simple motor task across 2 runs. In the task, participants were asked to move an orange dot to specified target locations on a semicircular rating scale using an MR-compatible joystick (**S5A Fig**). This task, consisting of 21 trials per run, was administered prior to the first and fourth runs of the pain prediction task. Target positions were randomly selected within 5 intervals, each covering approximately one-fifth of the semicircle. Due to a data-saving problem, two-run data from 4 participants and one-run data from 15 participants were excluded. Thus, we analyzed the simple motor task data of the remaining 55 participants. We first conducted a correlation analysis between the target location and the final position of the dot (**S5B Fig**). The results showed that participants, on average, exhibited a high level of correlation (mean $r$ = 0.9425; one-sample $t$ test: $t(54)$ = 94.6784, SD = 0.0738, $p$ = 1.0511×10$^{-61}$), indicating precise task performance. Then, to examine the brain activation patterns associated with the simple motor task, we conducted an fMRI single-trial analysis using the same approach as with the pain prediction task data. We conducted robust regression with the averaged single-trial brain activation map for each participant [79] (see **S5C Fig** for the results).

## ROI-based temporal mediation analysis

We conducted a region-of-interest (ROI)-based approach to complement the results of the whole-brain temporal mediation analysis results. This approach allows us to examine the temporal dynamics of brain mediation for specific regions, minimizing the need for some exploratory approaches such as ICA. Based on prior literature [13,16,66], we selected 17 a priori ROIs known for their roles in pain processing. These include areas involved in nociceptive pain processing [66] (e.g., left and right thalamus, right second somatosensory area, and left and right dorsal-posterior insula cortex), regions associated with self-regulatory strategies in response to pain [16] (e.g., ventromedial prefrontal cortex and nucleus accumbens), and areas linked to social information processing related to pain [13] (e.g., pregenual anterior cingulate cortex, orbitofrontal cortex, left and right ventrolateral prefrontal cortex, left supramarginal gyrus, medial frontal gyrus, intraparietal sulcus, and right dorsolateral prefrontal cortex; see **S9A Fig**).

For the ROI-based temporal mediation analysis, we first obtained the average activity for each ROI across the 45 TRs from the single-trial FIR model. We then assessed the mediation effects of ROI activity at specific time points on continuous pain prediction ratings (spanning 32 TR bins). In our mediation model, cue conditions (coded as −1, 0, and 1) and stimulus intensity levels (coded as 1 to 5) were the predictors ($x$), ROI activity served as the mediator ($m$), and pain prediction ratings were the outcome variable ($y$). When cue condition served as a predictor, stimulus intensity was included as a covariate, and in the model where the stimulus intensity served as a predictor, the cue levels were included as a covariate.

To test the significance of the mediation effects, we employed bootstrap tests with 10,000 iterations. This process resulted in 45×32 result matrix for each ROI, similar to the approach taken in the whole-brain analysis. To identify the significant temporal domains within each ROI, we used the "spm_bwlabel.m" function in SPM12 (Wellcome Trust Centre for Neuroimaging). This function detects connected multiple clusters within a 2D temporal domain result matrix after applying a threshold of $q < 0.05$, FDR corrected [73]. The results of this analysis are shown in **S9B and S9C Fig**.

## Supporting information

**S1 Data.** Data for generating Figs 1C (overall pain ratings by condition per participant), 1E (GLM results by time point), 4D (proportions of the mediation maps across principal gradient bins), 5D (proportions of the mediation maps across principal gradient bins), S1 (the results of the pain calibration task), S5B (the trial-by-trial target and response position per participant), S6 (the correlation coefficients between continuous pain prediction ratings at each time and overall pain ratings over time), and S13 (the results of principal gradient analysis with our own resting-state fMRI data).
(ZIP)

**S1 Fig. Results of pain calibration task. (A)** The pain calibration task was conducted to determine individualized stimulus intensity levels to match the levels of subjective pain experience across individuals. Each dot represents the temperature calibrated for an individual, corresponding to each stimulus intensity level. The mean temperatures (standard deviation) for each stimulus level are as follows: 44.49˚C (1.34) for LV1, 45.25˚C (1.30) for LV2, 46.02˚C (1.27) for LV3, 46.78˚C (1.29) for LV4, and 47.47˚C (1.26) for LV5. One-way analysis of variance results revealed a significant main effect of the fitted stimulus intensity, $F(4, 290) = 49.02$, $p = 1.786e{-}31$. **(B)** The distribution of the $R^2$ coefficients of the final linear regression models from the pain calibration task. **(C)** Each line represents the range of calibrated stimulus

intensities for each participant, and dots at the end of lines indicate LV1 and LV5 stimulus intensity. The order of participants is sorted by the temperature of LV1. Data from 59 participants who passed the pain calibration task were displayed in this figure. The underlying data for S1 Fig can be found in S1 Data.
(DOCX)

**S2 Fig. Average trajectory of continuous pain prediction ratings for all experimental conditions.** Thick and solid lines represent the group average of rating trajectory, while dotted lines represent the within-subject standard error of the mean (SEM). Thin lines illustrate individual averages of rating trajectories. Different colors correspond to experimental conditions.
(DOCX)

**S3 Fig. Conjunction of the cue and the stimulus mediation maps.** The conjunction map shows the spatial overlap and unique areas involved in the cue and stimulus mediation. Regions colored in cyan and pink denote brain areas associated with the mediation of cue and stimulus effects, respectively. These are thresholded at voxel-wise FDR $q < 0.05$ with cluster extent $k > 5$. The areas where these 2 mediation maps overlap are colored in purple.
(DOCX)

**S4 Fig. Independent component analysis scheme for the temporal mediation analysis results. (A)** A schematic diagram that illustrates the step-by-step procedure to identify the temporal domains and brain mediators through independent component analysis (ICA). The temporal mediation result matrices ($p$-values for Path $a \times b$) served as the input for the ICA. We eliminated uninterpretable time domains from analyses, including the mediation effect of later brain activity on earlier ratings or the influence of stimulus intensity on ratings preceding thermal stimulation. These excluded time domains appear as blank in the result matrices. To make the input matrix, we first vectorized the mediation result matrices (i.e., $p$-values for Path $a \times b$) for each voxel and converted them into -log10($p$). Next, we concatenated these vectors across all voxels, thereby aggregating the data from the whole brain. This resulted in matrices with dimensions of 880 or 602 by 24,860. We then obtained temporal and spatial component weights using the GIFToolbox with the fastICA algorithm (# of components = 5). The analysis yielded 2 sets of 5 components, one for spatial weights (24,860 [# of voxels] × 5 [# of components]) and the other for temporal weights (880 or 602 [# of temporal mapping] × 5 [# of components]). To identify the brain mediators and relevant temporal domains for each component, we first thresholded the temporal weights with the top 2.5 percentile. The resulting 5 thresholded temporal weights could be described using a river plot connecting the brain activity and pain ratings. We then identified brain mediators related to these temporal domains based on the following 3 criteria: (1) voxels that survived the FDR correction for multiple comparisons at $q < 0.05$; (2) regions with at least 5 contiguous voxels; and (3) the survived voxels should cover at least 5% of the defined temporal domain. **(B, C)** The matrices and brain maps represent the spatial weights and temporal components from the ICA analysis. Warm vs. cool colors represent high vs. low component weights. The areas enclosed by the red outlines represent the top 2.5% of the weights. One of the identified components was excluded from further analysis due to its strong association with the visual network (right panel), potentially reflecting task-related processes.
(DOCX)

**S5 Fig. Simple motor task. (A)** Participants completed a simple motor task over 2 runs. They were instructed to move an orange dot to specified target locations on a semicircular rating scale using an MR-compatible joystick. Each run consisted of 21 trials, with each trial including an inter-trial interval of 3–7 s and 6 s for moving the dot. **(B)** Correlations between the

target and actual final positions of the orange dot. Each regression line represents the relationship between the target and actual final positions for individual participants. The angle of 0 to 180˚ was normalized to a 0 to 1 scale. **(C)** (Left) Brain maps displaying activation patterns during the simple motor task (FDR $q < 0.0001$ and a cluster size $k < 2$ voxels). (Right) Common brain mediators for both cue and stimulus intensity from **Fig 6B**. Red dashed circles highlight the somatomotor and visual cortex regions that served as common brain mediators for both cue and stimulus effects. The underlying data for S5B Fig can be found in S1 Data.
(DOCX)

**S6 Fig. Correlations between the continuous pain prediction ratings and the overall pain ratings.** To investigate the relationship between continuous pain prediction ratings at each time window (comprising 32 TR-level bins) and overall pain ratings, we averaged the continuous pain ratings within each TR window. We then calculated Pearson's correlation between these time-averaged continuous ratings and the overall pain ratings. The resulting plot demonstrates an overall increase in correlations throughout the trial. Analysis of variance (ANOVA) results revealed a significant time effect, $F(29, 1{,}740) = 62.35$, $p = 1.60276\text{e-}244$. This finding implies that the later stages of the continuous pain prediction ratings provide more information into the overall pain ratings compared to the earlier and middle stages. The underlying data for S6 Fig can be found in S1 Data.
(DOCX)

**S7 Fig. Signs of Paths *a* and *b* of brain mediators for cue and stimulus intensity effects.** Each row of brain maps corresponds to temporal domains of the **(A)** cue and **(B)** stimulus mediation. Brain regions where Paths *a* and *b* are both positive are depicted in red, and regions where both Paths *a* and *b* are negative are depicted in blue. Radial plots show the relative proportions of the number of overlapping voxels between the thresholded mediation maps with positive (red) and negative (blue) Paths *a* and *b* and each of the large-scale networks (or regions) given the total number of voxels within each network (or region). VA, ventral attention network; DA, dorsal attention network; SM, somatomotor network; BS, brainstem; HC/Amy, hippocampus and amygdala; TH, Thalamus; DM, default mode network; FP, frontoparietal network; LIM, limbic network.
(DOCX)

**S8 Fig. Temporal mediation analysis with different hyperparameters.** To assess the robustness of the temporal mediation analysis results presented in **Figs 4** and 5, we performed the same analysis with different hyperparameters and evaluated the spatial similarity with the original mediation maps. **(A)** Analysis Pipeline (left): We obtained brain mediation maps using the same analysis pipeline from the original results, modifying only 1 hyperparameter. We then calculated the spatial similarity between thresholded maps using Dice coefficients. The spatial similarity was visualized via multidimensional scaling (MDS) in 2 dimensions. In the resulting MDS plots, the distance between brain mediators represents their similarity, with greater similarity indicated by closer proximity. Example results (middle): For instance, testing a new threshold for temporal weights yields 5 new mediation maps, one for each component. We can compute the spatial similarity among 10 mediation maps—5 from the original results and 5 from the new analysis results. Focusing on the orange box cases, we examine the relationship between the first mediation map from the original results and the first and fifth mediation maps from the new analysis results. A shorter distance in the MDS plot reflects greater similarity between the original and first new mediation maps, while a larger distance indicates lower similarity between the original and fifth new mediation maps. Figure legend for **(C)** (right): The color scheme used in **Figs 4** and 5 was applied to represent mediation maps for

cue and stimulus intensity. The dark gray dot signifies a mediation map from the visual cortex-dominant component. The numbers in the MDS plots denote new mediation maps derived from analyses with varying hyperparameters. **(B)** We varied the number of ICA components, the percentile of temporal component weights, and the coverage of the defined temporal domain for voxel thresholding one at a time, while keeping all other analysis parameters consistent with **Figs 4** and **5**. **(C)** Spatial similarity results: MDS was used to project the spatial similarities between brain mediators from the main and additional analyses into a 2D space. Colored circles represent the mediators from the original results in **Figs 4** and **5**, while number markers correspond to the results from new analyses with different hyperparameters. Mediation maps with altered temporal weights and voxel coverages generally aligned with the original results. Even when more components were tested, many new brain mediation maps exhibited high similarity to the original maps. Overall, the additional analyses with varying hyperparameters produced brain mediator patterns largely consistent with the original findings.
(DOCX)

**S9 Fig. Region-of-interest (ROI)-based temporal mediation analysis results.** We conducted a region-of-interest (ROI)-based approach to complement the results of the whole-brain temporal mediation analysis results. For a detailed description of the ROI-based temporal mediation analysis, please refer to the Materials and methods. **(A)** Based on prior literature [13,16,66], we selected 17 a priori ROIs known for their roles in pain processing. These include areas involved in nociceptive pain processing [66] (e.g., left and right thalamus, right second somatosensory area, and left and right dorsal-posterior insula cortex), regions associated with self-regulatory strategies in response to pain [16] (e.g., ventromedial prefrontal cortex and nucleus accumbens), and areas linked to social information processing related to pain [13] (e.g., pregenual anterior cingulate cortex, orbitofrontal cortex, left and right ventrolateral prefrontal cortex, left supramarginal gyrus, medial frontal gyrus, intraparietal sulcus, and right dorsolateral prefrontal cortex). **(B, C)** River plots illustrate the ROI-based temporal mediation analysis results. The results were categorized by brain mediation timing—"early" and "late." In addition, the regions with significance in more than 2 domains were categorized as "multiple." The plots show the temporal domains significant at a false discovery rate (FDR) $q < 0.05$, alongside adjacent time domains pruned using 2 more liberal thresholds ($p < 0.005$ and $p < 0.01$) to contextualize the results. FDR, false discovery rate; vmPFC, ventromedial prefrontal cortex; pgACC, pregenual anterior cingulate cortex; NAc, nucleus accumbens; dpINS, dorsal-posterior insular; S2, second somatosensory area; OFC, orbitofrontal cortex; vlPFC, ventrolateral prefrontal cortex; SMG, supramarginal gyrus; MFG, medial frontal gyrus; IPS, intraparietal sulcus; dlPFC, dorsolateral prefrontal cortex.
(DOCX)

**S10 Fig. The actual slide presented on Days 1 and 2 to ensure participants fully understood the rating scale and the pain prediction rating. (A)** Participants were thoroughly instructed on how to use the scale before proceeding to the pain calibration task on Day 1. For this, we used a structured instruction script, and the instructions focused primarily on the anchors and corresponding labels of the scale. As noted in the main text, we employed a modified version of the generalized labeled magnitude scale (gLMS [70]) to define the anchors. These anchors included no sensation (0˚), weak (18˚), moderate (50˚), strong (108˚), very strong (162˚), and strongest imaginable sensation (180˚). We emphasized that the label "Weak" marked the point where the sensation began to feel painful. To ensure participants fully understood the instructions, they were asked to explain the scale back to experimenters. These figures show the slide presented for this "explaining-back" procedure. If a participant was unable to explain the scale

correctly, we provided the instructions again until they fully grasped the scale. **(B)** The key aspect of the prediction rating was for participants to continuously report their ongoing estimate of how painful the stimulus would be, indicating their "current" prediction of the pain level. (Top) We showed participants an example of continuous pain prediction ratings to explain the pain prediction rating and how to report it. We instructed them the angle from the start point will be regarded as a current pain prediction. We also instructed that they did not need to click the button to report the prediction continuously, highlighting the difference between the prediction ratings and the intensity ratings. (Bottom) We asked participants to explain what they understood about the pain prediction back to the experimenter. If their understanding was incomplete, the procedure was repeated until they fully grasped it. The English translation of the text was not displayed during the actual experiment.
(DOCX)

**S11 Fig. Trial structure and rating trajectory examples. (A)** Schematic overview of trial structure. **(B)** Example of the continuous pain prediction rating: (Left) The figure shows an example rating trajectory over the 14.5 s during the pain prediction rating period. Starting from the center (i.e., the initial point), participants were asked to move the orange dot anywhere within the screen to report their continuous ratings. We emphasized that it was the angle, not the distance, that reflected participants' rating, and that they were not required to reach the outer end of the semicircles. The rating trajectory was recorded with the dot's *x* and *y* coordinates, but it was converted to the angle from the left segment of the semicircle base. (Right) The (*x*, *y*) coordinates were converted to the angles, which served as online ratings. Only the current dot location represented as the orange dot was visible to participants. **(C)** Example of the overall pain rating: (Left) The figure shows an example rating trajectory for the overall pain rating. (Right) The dot's (*x*, *y*) coordinates were converted to the angles. Then, we calculated the average of the angles of the last 1 s, which served as the overall pain rating. We made a short video of an example trial (https://youtu.be/tR0LUDaYt38) to illustrate what participants saw in the scanner.
(DOCX)

**S12 Fig. Temporal signal-to-noise ratio (tSNR) map.** We calculated the temporal signal-to-noise ratio (tSNR) using the TR-level data of the prediction task runs. We then averaged the tSNR values across runs and participants. The map shows the group average of the tSNR values.
(DOCX)

**S13 Fig. Building a volumetric gradient map.** Principal gradient map diagnostics and resulting map. **(A)** Scaled Eigenvalues across components. **(B)** A scatter plot of the first and second connectivity-based principal gradients. Each dot in the scatter plot represents connectivity, and different color represents the relationship between the first and second gradients. The gradient pattern was largely consistent with the original study [33]. **(C)** A brain map representing the first and second gradients using the same color scheme as the scatter plot in **B**. The underlying data for S13 Fig can be found in S1 Data.
(DOCX)

**S14 Fig. A large-scale functional brain networks and subcortical regions.** Different colors represent different large-scale networks and subcortical regions. A large-scale functional network includes seven cortical [75], basal ganglia [77], and cerebellum networks [76]. In addition to these large-scale networks, we added thalamus, hippocampus/amygdala, and brainstem [78].
(DOCX)

## Acknowledgments

We thank Taenyun Kim, SeongJae Park, Minie Jung, Jinwon Park, Hong Ji Kim, and Soo Ahn Lee for their help in conducting the experiment.

## Author Contributions

**Conceptualization:** Suhwan Gim, Choong-Wan Woo.

**Formal analysis:** Suhwan Gim, Choong-Wan Woo.

**Funding acquisition:** Suhwan Gim, Choong-Wan Woo.

**Investigation:** Suhwan Gim, Choong-Wan Woo.

**Methodology:** Suhwan Gim, Choong-Wan Woo.

**Supervision:** Choong-Wan Woo.

**Visualization:** Suhwan Gim, Choong-Wan Woo.

**Writing – original draft:** Suhwan Gim, Choong-Wan Woo.

**Writing – review & editing:** Suhwan Gim, Seok-Jun Hong, Elizabeth A. Reynolds Losin, Choong-Wan Woo.

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
