## [Editor Report · Decision Letter 0]

20 May 2024

Dear Dr Woo, 

Thank you for submitting your manuscript entitled "Temporal Dynamics of Brain Mediation in Predictive Cue-induced Pain Modulation" for consideration as a Research Article by PLOS Biology.

Your manuscript has now been evaluated by the PLOS Biology editorial staff as well as by an academic editor with relevant expertise and I am writing to let you know that we would like to send your submission out for external peer review.

Once your full submission is complete, your paper will undergo a series of checks in preparation for peer review. After your manuscript has passed the checks it will be sent out for review. To provide the metadata for your submission, please Login to Editorial Manager (https://www.editorialmanager.com/pbiology) within two working days, i.e. by May 22 2024 11:59PM.

Kind regards,

Luke

Lucas Smith, Ph.D.

Senior Editor

PLOS Biology

lsmith@plos.org

---

## [Decision Letter · Decision Letter 1]

3 Jul 2024

Dear Dr Woo,

Thank you for your patience while your manuscript "Temporal Dynamics of Brain Mediation in Predictive Cue-induced Pain Modulation" went through peer-review at PLOS Biology. Your manuscript has now been evaluated by the PLOS Biology editors, an Academic Editor with relevant expertise, and by several independent reviewers.

In light of the reviews, which you will find at the end of this email, we are pleased to offer you the opportunity to address the comments from the reviewers in a revision that we anticipate should not take you very long. We will then assess your revised manuscript and your response to the reviewers' comments with our Academic Editor aiming to avoid further rounds of peer-review, although might need to consult with the reviewers, depending on the nature of the revisions.

**IMPORTANT - SUBMITTING YOUR REVISION**

*Resubmission Checklist*

*Published Peer Review*

*PLOS Data Policy*

*Blot and Gel Data Policy*

Sincerely,

Christian Schnell (on behalf of Lucas)

Christian Schnell

Senior Editor

PLOS Biology

cschnell@plos.org

Lucas Smith, Ph.D.

Senior Editor

PLOS Biology

lsmith@plos.org

REVIEWS:

Reviewer #1 (Tamas Spisak): This manuscript presents a well-designed and executed study employing innovative techniques, such as continuous rating trajectories and temporal mediation analysis, to provide fresh insights into the neural mechanisms underlying the temporal dynamics of pain modulation. The study's presentation is excellent, and it provides with a wealth of valuable information, even in the supplementary material. With minor revisions, I believe it is suitable for publication in PLOS Biology.

My most important remark is the apparent overabundance of "free" parameters in the temporal mediation analysis (e.g., number of components, top percentile of temporal weights, criteria for selecting brain mediators, etc). This raises concerns about the robustness of the otherwise very exciting temporal mediation results. A supplementary analysis of the robustness of the results to these parameters would be appreciated.

In terms of the presentation, I agree that it is appealing to discuss the results in terms of a continuous Bayesian updating process. However, the study provides limited evidence that the dynamic changes in the mediation pattern are directly related to such processes. There is no direct evidence that priors are updated. If we adhere to the Bayesian theoretical framework, the results can also be explained by an increasingly precise sensory likelihood component during the continuous Bayesian integration process. (And I am not sure these results need to be forced into this framework at all.) The relevant parts in the text should be improved by acknowledging these limitations or even toned down (in the abstract, introduction, and discussion).

The manuscript would also benefit from improved motivation and description of the semicircular layout. The motivation should be introduced earlier in the text, rather than leaving the reader to discover in the methods section that the main reason is to address the issue of intensity rating being confounded with cursor movement distance by making the starting point equidistant from all possible ratings. Actually, I remain somewhat unconvinced that the semicircular layout does not induce bias in the ratings. As the distance from the initial point increases, the same angle might appear more extreme to the participant compared to the center line. Furthermore, if I understood correctly, participants saw their own rating history within a block, which could induce a "self-reinforcement effect" where participants strive to remain consistent with their previous ratings.

Other minor remarks/questions:

 - A brief description of the paradigm earlier in the text would be useful. Initially, it is unclear how pain stimuli changed during the experiment. More information is needed in the introduction for readability.

 - Some parts of the introduction are somewhat redundant (last two paragraphs).

 - "Are the temporal mediation findings sensible and interpretable in terms of the large-scale functional organization of the brain?" It's a matter of taste, of course, but this might be phrased too broad to be a meaningful research question.

 - A short, precise description of what mediation means statistically would benefit the reader.

 - While the temporal mediation analysis is well explained in the methods and Figure S4 is also very-very helpful, more clarity in the Results section would be beneficial.

 - Why does the visual cortex mediate late ratings so much? What did the participants see in the late phase? Could it be a self-reinforcing feedback from their own ratings (confirmation bias)?

 - Do we see path a or b effects or mediation effects in any motor areas, possibly related to the continuous rating?

Reviewer #2 (Francesca Fardo): This study presents novel findings on the temporal dynamics of cue-induced pain modulation using fMRI. The methodological approaches include a sophisticated continuous two-dimensional scale to rate predictions about upcoming pain, temporal mediation analyses to investigate the integration of predictions with sensory inputs over time, and principal gradient mapping to interpret the hierarchical integration from transmodal to unimodal brain areas in cue-induced pain modulation. These innovative and rigorous methods effectively address the research questions posed.

The statistical analyses are robust and appropriately conducted, utilizing multilevel general linear models (GLM) and bootstrapping to ensure the reliability of the results. The supplementary information provided is comprehensive and supports the main findings of the study. Additionally, the data and code availability statements reflect the authors' commitment to transparency and reproducibility. I believe that the supplementary figures and methodological details are sufficient to enable replication of the study.

I commend the authors for the thoroughness, complexity, and clarity of their manuscript. The figures are elegant and visually pleasing, clearly reflecting the significant effort and dedication invested in this work. Overall, this manuscript makes a significant contribution to the field, and the comments below can be addressed with textual changes in the manuscript.

Comments:

- The use of a semicircular display for assessing prediction ratings seems sub-optimal due to the unequal distribution of space available for participants. This design results in smaller spaces towards the center and larger spaces towards the periphery, leading to unequal variability in rating resolution throughout the task. Additionally, the semicircular shape introduces a non-linear scaling of pain predictions, where small angular changes near the center represent different levels of intensity change compared to the same angular changes near the periphery. Although motivated by the need to set the starting point equidistant from all other points, this design choice introduces complexity and potential confusion for participants, as the use of angle might not be intuitive. These issues should be acknowledged in the limitations section.

- Although the semicircular scale aims to reduce the impact of movement-related confounds by using angle rather than distance, there might still be motor confounds related to how participants move the joystick. Movements towards the outer edges might require different motor control compared to movements towards the center. It would be beneficial to clarify what motion corrections were included as regressors in the GLM models to address these potential confounds. Additionally, more information on how the "simple motor task" was used in the analyses and interpretation of the results would be important to include.

- The current discussion of the transition from transmodal to unimodal brain areas does not adequately recognize the circularity of brain communications. One would expect continuous, cyclical interactions between transmodal and unimodal areas. Acknowledging and discussing this continuous interaction would enhance the manuscript's explanation of the brain's integrative processes in pain perception.

- The Declaration of Helsinki should be mentioned to ensure ethical standards are addressed.

- The phrase "write an informed consent" should be corrected to "sign the informed consent."

Reviewer #3: The manuscript by Gim and colleagues was a pleasure to read, and I would like to congratulate the authors on an excellent study. There are many things to like here: the manuscript is very well written, the figures adequately convey the (quite complex) sets of results in a beautiful manner, the sample size is on the large side for typical fMRI experiments, and the data are rigorously analyzed. On top of this, the authors introduce two key novel methodologies (semicircular rating scale and temporal mediation analysis) that will be highly beneficial to pain research in the future and based on which they have constructed a very elegant paradigm to look at the dynamics of integrative processes in pain perception. Overall, I strongly recommend publication of this work and only have a few points that would need to be addressed. 

1. There are two aspects of the design that are currently unclear to me and where I believe it could be helpful if the authors were to explain the reasoning behind their choices. First, why did the thermal stimulation use such a slow ramp (5s to target temperature), considering that the chosen thermode is capable of faster ramping-behaviour to my knowledge? Second, why did they employ such a long and variable delay between cue offset and pain onset? 

2. While a real strength of the manuscript is the development of the semicircular rating scale (providing several advantages over a VAS), there are a few issues the authors should address. First, how do we know how participants interpret the scale, considering that it ranges from 'no sensation' to 'strongest imaginable sensation' and thus confounds innocuous temperature changes (close to the left end of the scale) with noxious ones (close to the right end of the scale)? In other words, where on the scale is the change from innocuous to painful percept occurring and will this be identical across participants? Second, how does the scale look, considering that different numbers of lines are used throughout the figures? Third, to make this novel tool useful for the pain research community, the authors should make the code available upon which the scale is based. Fourth, how do we know that participants indeed use the scale as intended, i.e. making pain 'predictions' instead of reporting pain 'intensity' (especially in later parts of the trial)?

3. There are several analysis choices where I am not able to understand the authors reasoning / approach. First, I am not able to follow why a different coding of predictors was used in the GLMs for overall pain ratings and continuous pain ratings (especially considering that cue weighting order was changed); this is even more relevant, considering that the authors go back to their original coding scheme when using the ROI-mediation analysis. Second, I am not able to understand how the design matrix looked for their fMRI single trial analysis: did they have one large design matrix where each trial was modelled separately or did they construct several design matrices (one for each trial, with the remaining trials in one regressor)? Third, to my knowledge in SPM's FIR approach there is no convolution with the HRF - why did the authors use such a combination of basis functions? Fourth, I cannot follow when, how and why they used temporal smoothing during the temporal mediation analysis - was this for regularization purposes and if so, on which data was it performed (fMRI, ratings, both)? Most importantly, why was the ICA limited to five components - were other dimensionalities explored (if so, what were the outcomes; if not, why focus on exactly five components)? While it is nice to see that the authors mention this limitation in the Discussion, I believe it is important to address this analytically (if feasible), considering that it strongly shapes the outcome of the temporal mediation.

Minor points

Based upon which reasoning did the authors decide on a sample size of 84 (completing first session) or 59 (completing entire experiment)?

The second paragraph describing the 'Pain calibration task' is very hard to follow - I would suggest the authors try to make their approach clearer here. It would also be helpful if the authors could explain in more detail how the 'pain prediction task with eight runs' worked during the structural data acquisition. 

Why were the two motor task runs acquired (I did not see any results based on these runs, though they could be informative for example with respect to sensorimotor cortex activation considering the ongoing rating)?

What was the purpose of the 'other-pain question' and where do the authors report the data from this question?

A multi-band factor of 8 is quite aggressive - did the authors consider possible slice-leakage artefacts? Also, was there no in-plane acceleration via GRAPPA (if not, how strong were distortions of the EPI data) and no partial-Fourier employed? Considering their very fast TR and low flip-angle, could they report tSNR measures in a supplementary figure?

The description of how the gradient map was created and what it means is not detailed enough - please consider expanding (also the legend of the associated figure), as currently it is difficult to follow and understand how this relates to their main findings. 

Why did they choose three studies from one group to inform their ROI-based mediation analysis - might this lead to biased insights?

---

## [Editor Report · Decision Letter 2]

2 Oct 2024

Dear Dr Woo,

Thank you for your patience while we considered your revised manuscript "Temporal Dynamics of Brain Mediation in Predictive Cue-induced Pain Modulation" for publication as a Research Article at PLOS Biology. This revised version of your manuscript has been evaluated by the PLOS Biology editors and the Academic Editor.

Based on our Academic Editor's assessment of your revision, we are likely to accept this manuscript for publication, provided you satisfactorily address the following data and other policy-related requests:

1) Please change your title to: “Spatiotemporal integration of contextual and sensory information within the cortical hierarchy in human pain experience”

2) Please include the approval number from the institutional review in your ethics statement, as well as the name of the institutional review board.

3) As per our Data policy, we require that all data underlying the figures is available. Please provide the data underlying 1C,E, 4D, 5D, S1A,B,C, S5B, S6, S13A,B. Please provide these in a file called S1_data. Please also check whether the error bars are present in figure 1C.

4) Ensure all used code is publicly available and citable.

5) Please note that per journal policy, the model system/species studied should be clearly stated in the abstract of your manuscript. Please note in the abstract that this is a human study.

6) We noticed that you have supplemental text and supplemental references. PLOS Biology doesn’t have a word count, or reference limitations, please incorporate this into the main manuscript.

We expect to receive your revised manuscript within two weeks. 

*Published Peer Review History*

*Press*

Sincerely,

Suzanne

Suzanne De Bruijn, PhD, 

Associate Editor

sbruijn@plos.org

PLOS Biology

ETHICS STATEMENT:

-- Please include the full name of the IACUC/ethics committee that reviewed and approved the animal care and use protocol/permit/project license. Please also include an approval number.

-- Please include information about the form of consent (written/oral) given for research involving human participants. All research involving human participants must have been approved by the authors' Institutional Review Board (IRB) or an equivalent committee, and must have been conducted according to the principles expressed in the Declaration of Helsinki.

DATA POLICY:

[Figs….]

CODE POLICY

Per journal policy, if you have generated any custom code during the course of this investigation, please make it available without restrictions. Please ensure that the code is sufficiently well documented and reusable, and that your Data Statement in the Editorial Manager submission system accurately describes where your code can be found. [IF APPLICABLE: As the code that you have generated to XXX is important to support the conclusions of your manuscript, its deposition is required for acceptance.]

---

## [Editor Report · Decision Letter 3]

16 Oct 2024

Dear Dr Woo,

Thank you for your patience while we considered your revised manuscript "Spatiotemporal integration of contextual and sensory information within the cortical hierarchy in human pain experience" for publication as a Research Article at PLOS Biology. This revised version of your manuscript has been evaluated by the PLOS Biology editors.

I want to thank you for all your work addressing our remaining concerns. However, we have a few remaining points:

1) checking all data in S1_Data, I realized that the data underlying figure 1C are the average data. Could you provide the raw data here? in addition, Can you ensure these are annotate these data, to ensure these are properly described?

2) we appreciate your thorough Data and code availability statement. However, could you add this to EM, and remove it from the manuscript?

3) Could you please describe in the legends where the data in the respective figures can be found? for instance, for the figures where we asked the underlying data, state something like 'underlying data can be found in S1_Data. My apology that this request was not clear in my previous letter.

Finally, a very minor point: I realized that the files in the S1_Data Zip file have been uploaded as separate files. As you will need to upload a modified file for figure 1C, could you ensure to upload these data as a zip file, to ensure it shows up as a zip file in our system.

I apologize for the inconvenience of my additional queries.

We expect to receive your revised manuscript within two weeks. 

*Published Peer Review History*

*Press*

Sincerely,

Suzanne

Suzanne De Bruijn, PhD, 

Associate Editor

sbruijn@plos.org

PLOS Biology

---

## [Editor Report · Decision Letter 4]

22 Oct 2024

Dear Dr Woo,

Thank you for the submission of your revised Research Article "Spatiotemporal integration of contextual and sensory information within the cortical hierarchy in human pain experience" for publication in PLOS Biology. On behalf of my colleagues and the Academic Editor, Markus Ploner, I am pleased to say that we can in principle accept your manuscript for publication, provided you address any remaining formatting and reporting issues. These will be detailed in an email you should receive within 2-3 business days from our colleagues in the journal operations team; no action is required from you until then. Please note that we will not be able to formally accept your manuscript and schedule it for publication until you have completed any requested changes.

PRESS

Sincerely, 

Suzanne De Bruijn, PhD 

Associate Editor

PLOS Biology

sbruijn@plos.org